

# Generalized sliding law applied to the surge dynamics of Shisper Glacier and constrained by timeseries correlation of optical satellite images

Flavien Beaud[1,2], Saif Aati[1], Ian Delaney[3,4], Surendra Adhikari[3], and Jean-Philippe Avouac[1]

[1]Division of Geological and Planetary Sciences, California Institute of Technology, Pasadena, CA, USA
[2]Now at Department of Geography, University of British Columbia, Vancouver, BC, CA
[3]Jet Propulsion Laboratory, California Institute of Technology, Pasadena, CA, USA
[4]Now at Institute of Earth Surface Dynamics, University of Lausanne, Lausanne, Switzerland

**Correspondence:** Flavien Beaud (flavien.beaud@ubc.ca)

**Abstract.**

Understanding fast ice flow is key to assess the future of glaciers. Fast ice flow is controlled by sliding at the bed, yet that sliding is poorly understood. A growing number of studies show that the relationship between sliding and basal shear stress transitions from an initially rate-strengthening behavior to a rate-independent or rate-weakening behavior. Studies that have

tested a glacier sliding law with data remain rare. Surging glaciers, as we show in this study, can be used as a natural laboratory to inform sliding laws because a single glacier shows extreme velocity variations at a sub-annual timescale. The present study has two parts: (1) we introduce a new workflow to produce velocity maps with a high spatio-temporal resolution from remote sensing data combining Sentinel-2 and Landsat 8 and use the results to describe the recent surge of Shisper glacier, and (2) we present a generalized sliding law and provide a first-order assessment of the sliding-law parameters using the remote sensing

dataset. The quality and spatio-temporal resolution of the velocity timeseries allow us to identify a gradual amplification of spring speed-up velocities in the two years leading up to the surge that started by the end of 2017. We also find that surface velocity patterns during the surge can be decomposed in three main phases, and each phase appears to be associated with hydraulic changes. Using this dataset, we are able to constrain the sliding law parameter range necessary to encompass the sliding behavior of Shisper glacier, before and during the surge. We document a transition from rate-strengthening to rate-

independent or rate-weakening behavior. A range of parameters is probably necessary to describe sliding at a single glacier. The approach used in this study could be applied to many other sites in order to better constrain glacier sliding in various climatic and geographic settings.

## 1 Introduction

Describing the physics of glacier basal motion is a core challenge in modern glaciology. Enhanced basal motion can lead to

the demise of large ice shelves, and would lead to rapid and significant sea level rise (e.g. Mouginot et al., 2019; Catania et al., 2020; Pattyn and Morlighem, 2020). Ice flow velocities exceeding one meter per day are considered 'fast', and correspond to an ice flow regime that is dominated by basal sliding rather than ice deformation. Fast ice flow at tidewater glaciers is





responsible for the majority of mass loss from the Greenland Ice Sheet (e.g. Mouginot et al., 2019) because the ice advected
into the ocean facilitates calving and melt. Glacier surges can show velocities reaching tens of kilometers per year (e.g. Hewitt,
1969; Meier and Post, 1969; Post, 1969; Kamb et al., 1985) which redistribute ice mass dramatically, and can lead to enhanced
ice loss, disturbed glacial runoff, and can create multiple hazards for local communities (e.g. Hewitt and Liu, 2010; Bhambri
et al., 2020). Fast ice flow velocities cannot be adequately explained by traditional Weertman-type sliding relationships (e.g.
Lliboutry, 1968; Iken, 1981; Tulaczyk et al., 2000; Iverson and Iverson, 2001; Schoof, 2005; Zoet and Iverson, 2015; Minchew
et al., 2016; Stearns and Van der Veen, 2018; Zoet and Iverson, 2020) that express the basal shear stress as proportional to the
sliding velocity $\tau_{\mathrm{b}} \propto u_{\mathrm{b}}^{1/n}$ (Weertman, 1972; Budd et al., 1979; Bindschadler, 1983, see also Fig. 1).

The idea that bed shear stress is bounded for large sliding velocities was proposed in early physical glaciology work (Lli-
boutry, 1968; Iken, 1981). Yet, this idea only started gaining more traction in the early 2000's (Tulaczyk et al., 2000; Iverson
and Iverson, 2001; Schoof, 2005). Since then, a growing body of research on the mechanics of glacier sliding has arisen (Tu-
laczyk et al., 2000; Iverson and Iverson, 2001; Schoof, 2005; Gagliardini et al., 2007; Iverson, 2010; Pimentel and Flowers,
2010; Iverson and Zoet, 2015; Tsai et al., 2015; Zoet and Iverson, 2015; Minchew et al., 2016; Joughin et al., 2019; Zoet
and Iverson, 2020), proposing new relationships between shear stress and sliding velocity. These relationships can generally
be partitioned into three regimes (Fig. 1; see also Minchew and Joughin, 2020): (1) Form drag, similarly to Weertman-type
relationships, where drag is dominated by viscous deformation of ice around bed obstacles, and shear stress increases mono-
tonically with sliding velocity (2) a transition regime, where shear stress approaches or even reaches its maximum value and
starts decreasing, and (3) skin drag, where drag becomes dominated by the friction between ice and the bed, and shear stress
reaches an asymptote (i.e. Coulomb failure, Tulaczyk et al., 2000; Iverson and Iverson, 2001; Iverson, 2010) or decreases
monotonically with sliding velocity (Fig. 1; Schoof, 2005; Gagliardini et al., 2007; Zoet and Iverson, 2015).

While these theories represent the physics of ice sliding better than their Weertman-type predecessors, they currently suffer
from a number of caveats that prevent testing them in the real world and using them in numerical simulations (Pimentel and
Flowers, 2010; Jay-Allemand et al., 2011; Beaud et al., 2014; Tsai et al., 2015; Joughin et al., 2019; Thøgersen et al., 2019):
(1) one has to choose between a slip relationship suited for a rigid un-deformable bed, i.e. bedrock, or for a deformable bed,
i.e. sediment, (2) tuning the parameters requires knowledge of small-scale bed properties (e.g. sediment characteristic, bed
geometry), and (3) the possible non-unique solutions renders numerical solving challenging (Pimentel and Flowers, 2010).

Surge-type glaciers undergo cyclic switches in their flow regime between slower-than-normal (quiescence) and faster-than-
normal (surge) velocities (Meier and Post, 1969). During quiescence, the glacier builds up ice mass in a reservoir area where
the resistance to ice flow exceeds the driving stress (Cuffey and Paterson, 2010). The surge occurs when the driving stress
exceeds the resistive stress and glacier velocities increase by typically an order of magnitude in comparison with quiescence
velocities. This local and dramatic velocity increase creates an ice bulge that travels down glacier as a kinematic wave (e.g.
Kamb et al., 1985; Mayer et al., 2011; Adhikari et al., 2017). If the ice mass redistribution is significant enough, the terminus
of the glacier will become the receiving area, thicken, and may start advancing. Surge-type glaciers only represent about 1%
of glaciers world-wide (Jiskoot et al., 2000), and appear to be concentrated in climatic clusters (e.g. Sevestre and Benn, 2015).





A recent theory for glacier surges suggests that a specific combination of climatic conditions is required to create an unstable equilibrium in surge-type glaciers, based on coupled mass and enthalpy budgets approach (Benn et al., 2019).

Surge-type glaciers can be used to assess how basal shear stress depends on sliding velocity. Surges are dynamic instabilities
during which surface velocities increase ten-fold, and have return periods typically between a decade and a century (Meier and Post, 1969; Clarke et al., 1986; Jiskoot et al., 2000; Jiskoot, 2011; Quincey et al., 2011; Sevestre and Benn, 2015). The back and forth oscillation between relatively low and high velocities for a single glacier makes it possible to test the temporal transition between different sliding regimes. Seasonal velocity fluctuations can also be used in principle to test friction laws, although their duration is relatively short (typicallly a few weeks) and their amplitude relatively low (typically 20–50% velocity
increase), thus requiring particularly high resolution and quality data.

Satellite imagery is a valuable source of information to study glacier dynamics. It allows monitoring with a large spatial coverage and metric ground resolution, including otherwise inaccessible areas (e.g. Burgess et al., 2013; Quincey et al., 2015; Armstrong et al., 2017; Gardner et al., 2019; Dehecq et al., 2019). The temporal resolution of remote sensing is however limited by the timespan between the acquisition of two images, often weeks to months, necessary to create a displacement
map. Glacier velocities derived from remote sensing will therefore consistently miss sub-monthly fluctuations, in particular those at the daily or sub-daily timescales which can be significant (e.g. Kamb et al., 1985; Iken and Bindschadler, 1986; Iken and Truffer, 1997). Advances in algorithms to model, register, and correlate optical images with a subpixel accuracy (e.g. COSI-Corr, MicMac, ASP, CIAS, Medicis; see Rupnik et al., 2017; Rosu et al., 2015; Beyer et al., 2017; Heid and Kääb, 2012), combined with the ever-growing volume of satellite remote sensing dataset available (e.g. Gardner et al., 2019), nonetheless
provides opportunities for glaciology studies. Remote sensing has thus offered insight into recent surges (e.g. Copland et al., 2011; Dunse et al., 2015; Round et al., 2017; Steiner et al., 2018; Chundley and Willis, 2019; Rashid et al., 2019; Bhambri et al., 2020; Paul, 2020), as well multi-annual glacier dynamics (Moon et al., 2014; Van de Wal et al., 2015; King et al., 2018) and basal sliding mechanisms (Minchew et al., 2016; Stearns and Van der Veen, 2018). In this paper, we apply image correlation to a time series of optical images using COSI-Corr (Leprince et al., 2007), which is accurate to 1/20th of the pixel size, and we use
a post-processing method based on principal component analysis (PCA) to measure spatio-temporal variations of ice velocity. This method is applied to openly-available Landsat 8 (L8) and Sentinel-2 (S2) optical images of two surge-type glaciers in the Pakistani Karakoram, Himalaya. We retrieve a timeseries of velocity maps with 60m pixels and a temporal resolution as low as 5 days over six-year from 2013 to 2019. We additionally use commercial optical images to create digital terrain models (DEMs) and constrain ice volume changes during the surge (Aati and Avouac, 2020).
Of the two neighboring surge-type glaciers, Shisper glacier (Fig.2) experienced a dramatic surge during the study period, which saw its terminus advance by $\sim$ 1.7 kilometers (Rashid et al., 2019; Bhambri et al., 2020). Over that period, the other glacier, Mochowar (Fig.2), appears to have remained stable. Given the lack of local weather data, we use Mochowar as a reference glacier that we assume reflects the ice flow response to local climate in absence of a surge.

This paper is composed of three parts. First, we show that rigid and deformable bed theories can be combined into a
generalized sliding relationship applicable for any glacier environment and dynamic evolution. Then, we present the remote sensing method, show the results for Shisper and Mochowar glaciers, and use the measurements to characterize the surge



of Shisper glacier. Finally, we use the data set collected for Shisper glacier to validate and constrain the generalized sliding relationship and contextualize how the generalized sliding relationship can improve our understanding of surge dynamics.

## 2 Generalized sliding relationship

The early work of Weertman (1957) relating glacier sliding, $u_{\mathrm{b}}$, over a rigid idealized bed to basal shear stress as $\tau_{\mathrm{b}} \propto u_{\mathrm{b}}$, set the standard for generations of glaciologists, and remains relevant (MacAyeal, 2019). This formulation, which was later modified to allow for a non-linear behavior, $\tau_{\mathrm{b}} \propto u_{\mathrm{b}}^{1/p}$ (Weertman, 1972), and for a dependence on effective pressure, $N$, $\tau_{\mathrm{b}} \propto (u_{\mathrm{b}} N)^{1/p}$ (Budd et al., 1979; Bindschadler, 1983) remains most widely used (e.g. Cuffey and Paterson, 2010). We will thereafter refer to this type of relationship as Weertman-type.

The bed of a glacier can only produce a finite amount of resistance to flow that is determined by its properties (e.g. Lliboutry, 1968; Iken, 1981; Tulaczyk et al., 2000; Iverson and Iverson, 2001; Schoof, 2005), challenging the validity of Weertman-type relationships which imply an unbounded resistance to flow. For bedrock, i.e. a rigid bed, that limit is dictated by the maximum slope of bedrock obstacle adverse to flow in contact with the ice (Lliboutry, 1968; Iken, 1981; Schoof, 2005; Gagliardini et al., 2007; Zoet and Iverson, 2015). For till, i.e. a deformable bed, that limit is dictated by the shear resistance of the material
(Tulaczyk et al., 2000; Iverson and Iverson, 2001; Iverson, 2010; Zoet and Iverson, 2020). As a consequence, past a velocity threshold, the shear stress tends to an asymptotic value or reaches a maximum and eventually decreases, while the sliding velocity increases (Fig. 1a). That decrease can be the result of bed-weakening feedbacks at high velocities. For a deformable bed, it could be associated with reworking of the till matrix and a change in its geotechnical properties (e.g. Clarke, 2005). For a rigid bed, the maximum adverse slope to flow can be engulfed in basal cavities as they grow (Fig. 1d and f). In short, the
relationship between basal sliding and shear stress is dominated by viscous drag as velocities remain below some threshold velocity. For sliding velocities slightly above that threshold there is a a transition regime, where viscous drag and skin friction are competing. Beyond that regime, skin friction becomes dominant and sliding can keep on increasing while shear stress reaches an asymptotic value or decreases (Fig. 1a, f and g).

The relationship between sliding and shear stress for rigid and deformable beds show similar patterns, yet are the result
of fundamentally different physical processes. We show that the two sliding relationships can be unified and expressed using a general glacier sliding relationship. The rigid-bed sliding relationship first proposed by Schoof (2005) and generalized by Gagliardini et al. (2007) is:

$$\tau_{\mathrm{b}} = CN \left( \frac{\chi}{1 + \alpha \chi^q} \right)^{1/p}, \tag{1}$$

where $\tau_{\mathrm{b}}$ is the basal shear stress, $C$ is a parameter that sets the maximum friction law value, i.e. that remains smaller or equal
to maximum slope of obstacles on a rigid bed (Gagliardini et al., 2007), $N = p_{\mathrm{ice}} - p_{\mathrm{w}}$ is the effective pressure at the glacier bed defined as the difference between overburden ice and water pressures, $\chi$ is a normalized velocity, $p$ is power law exponent, and $q$ is an empirical exponent that relates to the strain weakening of the bed. Note that $q$ should be greater than, or equal to 1

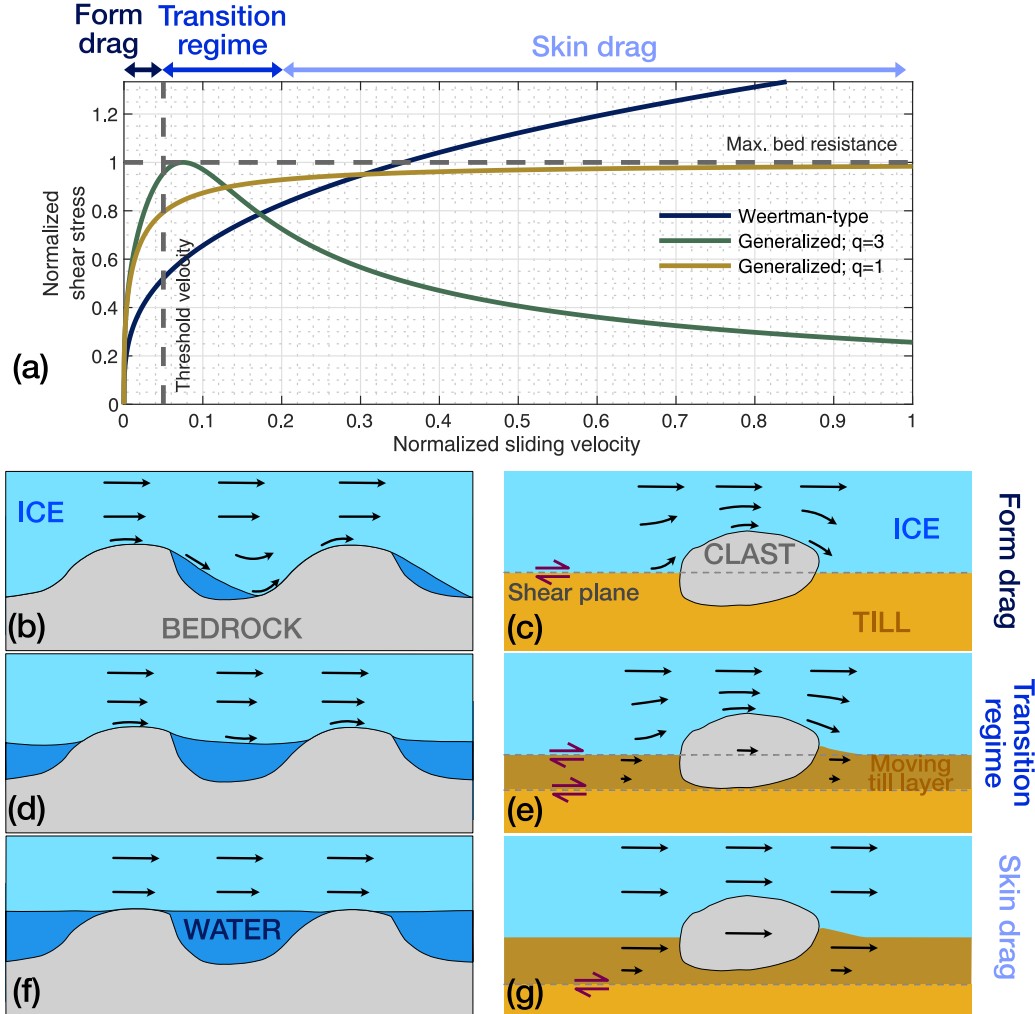

**Figure 1.** Conceptual representation of glacier sliding relationships for rigid and deformable beds, inspired by Minchew and Joughin (2020). (a) Plot of normalized shear stress versus normalized sliding velocity, comparing Weertman-type sliding and two forms of the generalized sliding relationship. (b)–(g) schematic representation of form drag (b, c), transition regime (d, e) and skin drag (f, g) for a rigid bedrock bed (b, d and f) and for a deformable till bed (c, e, and g). For a rigid bed, the skin friction regime can be likened to a curling stone sliding on an ice track, while it represents Coulomb failure for a deformable bed. Note that the shear stress is normalized by an imposed $\sigma_{\max} = 150 \times$ kPa, the sliding velocity normalized arbitrarily by $40 \, \mathrm{m \, day^{-1}}$, $u_{\mathrm{t}} = 2 \, \mathrm{m \, day^{-1}}$, and $p = 3$. For the Weertman-type relationship, we use $\tau_{\mathrm{b}} = (u_{\mathrm{b}} N / C_{\mathrm{s}})^{1/p}$, with $N \approx 600 \times$ kPa, and $C_{\mathrm{s}} = 2.5 \times 10^{-9} \, \mathrm{m \, a^{-1} \, Pa^{-p}}$.

and that the relationship is rate-independent if $q = 1$, and rate-weakening if $q > 1$. The term $\alpha$ is defined as $\alpha = \frac{(q-1)^{q-1}}{q^q}$ so that the maximum of $\tau_{\mathrm{b}}$ is $CN$, and $\chi$ is defined as


$$\chi = \frac{u_{\mathrm{b}}}{C^p N^p A_{\mathrm{s}}}. \tag{2}$$





The term $u_\mathrm{b}$ is the sliding velocity and $A_\mathrm{s}$ is a sliding parameter without cavity. Note that $C^p N^p A_\mathrm{s}$ has the dimension of a velocity. Zoet and Iverson (2020) propose the following sliding relationship for a deformable bed:

$$\tau_\mathrm{b} = N \tan(\phi) \left( \frac{u_\mathrm{b}}{u_\mathrm{b} + u_\mathrm{t}} \right)^{1/p}, \tag{3}$$

where $\phi$ is the friction angle of the till, and $u_\mathrm{t}$ is a threshold sliding velocity above which till resistance is defined by its
Coulomb strength. The sliding relationships for rigid (Eq. 1) and deformable (Eq. 3) beds can be reconciled by defining two bed-dependent variables. First, we define a maximum resistive stress $\sigma_\mathrm{max}$ as

$$\sigma_\mathrm{max} = \begin{cases} NC & \text{for a rigid bed} \\ N \tan(\phi) & \text{for a deformable bed,} \end{cases} \tag{4}$$

then we generalize the threshold velocity

$$u_\mathrm{t} = \begin{cases} C^p N^p A_\mathrm{s} & \text{for a rigid bed} \\ f(N, \text{bed properties}) & \text{for a deformable bed (see Eq. 2 in Zoet and Iverson, 2020),} \end{cases} \tag{5}$$

leading to the generalized sliding relationship:

$$\tau_\mathrm{b} = \sigma_\mathrm{max} \left( \frac{\frac{u_\mathrm{b}}{u_\mathrm{t}}}{1 + \alpha \left( \frac{u_\mathrm{b}}{u_\mathrm{t}} \right)^q} \right)^{1/p}. \tag{6}$$

Note that for a deformable bed, substituting Eqs. 4 and 5 into Eq. 6 and setting $q = 1$, simplifies to Eq. 3. We use the more general law expressed by Eq. 6 in our analysis.

## 3   Study area and data set

### 3.1   Study area

Shisper and Mochowar glaciers are located in the Hunza Valley, in the Northwestern Pakistani Karakoram (Fig. ,2). The Karakoram region shows a high concentration of surge-type glaciers (e.g. Hewitt, 1969; Sevestre and Benn, 2015), with approximately 90 such glaciers documented (Copland et al., 2011). A climatic anomaly allows glaciers in the Karakoram to maintain their ice volume (e.g. Hewitt, 2005; Gardelle et al., 2012; Kääb et al., 2015; Farinotti et al., 2020). Shisper and Mo-
chowar glaciers occupy adjacent valleys connecting as a Y and were merged until 2005. They are denominated Hassanabad glacier in the Randolph Glacier Inventory (Pfeffer et al., 2014). The relief in the area is dramatic as the elevation drop between Shisphare peak (7611 m a.s.l.) and Shisper glacier terminus ( 2500 m a.s.l.) is larger than 5000 meters over 15 kilometers. The glacier bergshrunds are perched between ∼5000 and ∼5500 m a.s.l. suggesting that the relief of each glacier is greater than 2000 meters. The glaciers we refer to as Shisper and Mochowar (Rashid et al., 2019), have also respectively been referred to
as Shispare and Muchuhar (Bhambri et al., 2020), or Shishper and Muchowar (Karim et al., 2020).

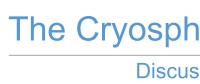
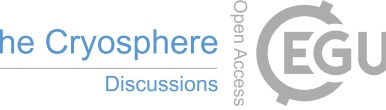

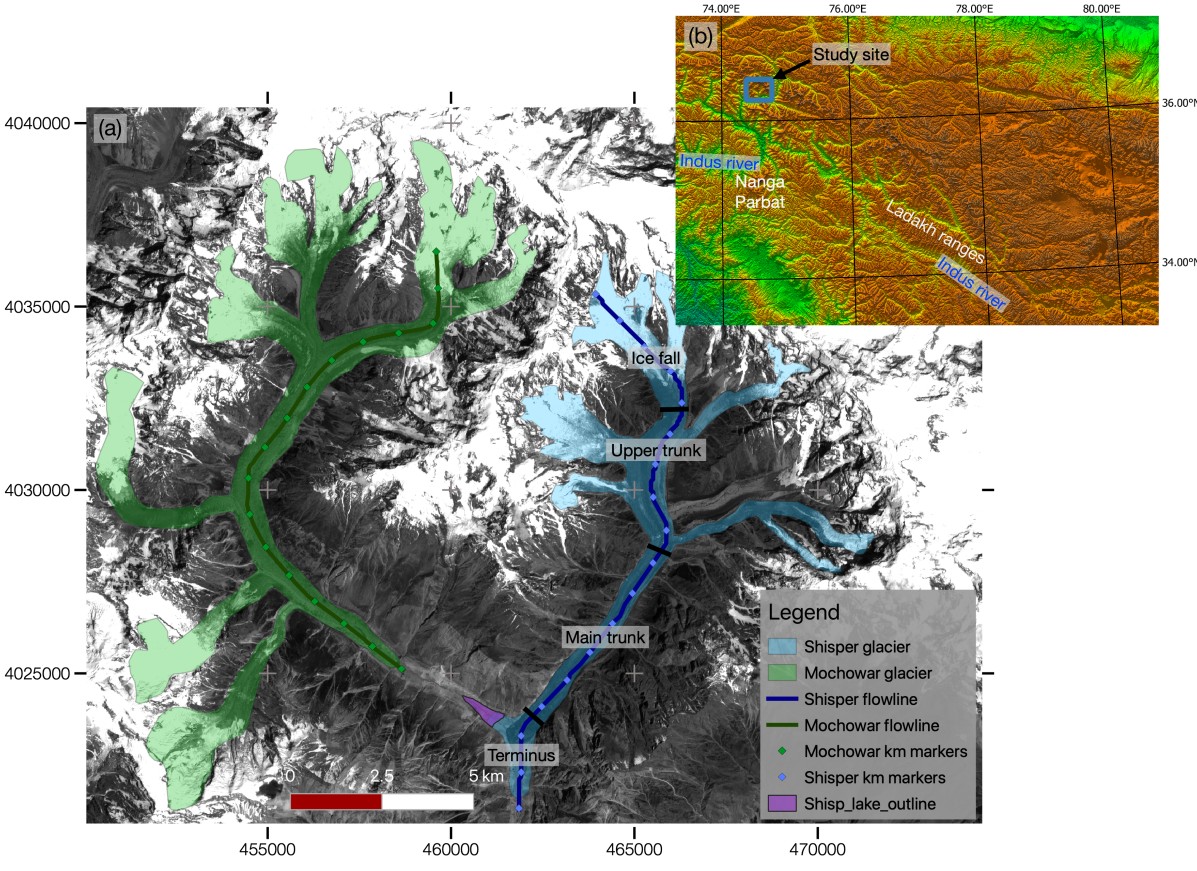

**Figure 2.** Overview of the study area. (a) Outlines of Mochowar (green) and Shisper (blue) glaciers with the location of the median flowlines used for analysis, and the maximum extent of the periglacial lake, overlaid on the Sentinel-2 optical image from 2018-07-15. The labels on Shisper glacier describe four zones identified as undergoing different dynamic processes during the surge. The ice fall zone spans across the accumulation and ablation area of Shisper glaciers where surface slopes are steep (up to 45°) and elevation > 3750 m a.s.l.. The upper trunk is within the ablation area and is where four tributaries merge with the main glacier. Surface elevations are comprised between 3750 m a.s.l. and 3400 m a.s.l. and the zone terminates just down-flow from the last identifiable ice fall along the profile. The main trunk is characterized by relatively uniform surface slope with no ice fall or tributary junction and surface elevations range from 3400 m a.s.l. and 2750 m a.s.l.. The terminus zone is below 2750 m a.s.l. and represent the area that show dynamic activity almost only linked to the surge bulge advancing. Coordinates are in meters and projected on zone 43 N of the Universal Transverse Mercator coordinate system. (b) Relief map of the Western Himalaya and Karakoram regions with notable geographic features for reference.

Bhambri et al. (2017) identified surges of Shisper in 1904–1905, 1972–1976 and 2000–2001, although inspection of Landsat images suggests that the last advance of Shisper glacier ended in 2005 (see section S2). A line of evidence indicating a past surge of Mochowar is the disintegration of 3.6 kilometers of its terminus in ~10 years, starting in 2005, suggesting it was stagnant ice left from a prior advance. Melt from both glaciers feed a hydroelectric power plant in the Hunza valley and



glacier-related hazards threaten the town of Hassanabad and the Karakoram Highway, which is the only paved road crossing
the mountain range (Shah et al., 2019; Rashid et al., 2019; Bhambri et al., 2020).

## 3.2   Data set

We use data from the Landsat 8 (L8; United States Geological Survey, USGS) and Sentinel-2 (S2; European Space Agency,
ESA) optical satellite systems to determine glacier surface velocities from May 2013 to August 2019. The two imaging systems
have similar spectral properties, ground resolution, acquisition rate (Table 1), and the data are free and open access (Roy et al.,
2014; Drusch et al., 2012). We utilize all images with less than 20% cloud cover over the study area, yielding a total of 100
images: 52 images acquired from L8 between the 5$^{th}$ of May, 2013 and 1$^{st}$ of April, 2019, and 48 images from S2 between the
21$^{st}$ of May, 2016 and 9$^{th}$ of August, 2019 (see Table S1).

Two satellites comprise the Sentinel-2 constellation, S2A and S2B. Both use the same Multi Spectral Instrument (MSI)
sensor (Table 1) and have a return time of 10 days, producing a staggered return interval of 5 days. S2B was launched later
than S2A, and the first usable image from that satellite is in November of 2017. Of the 48 S2 images, 25 images were acquired
from the S2A sensor and 23 images were taken from the S2B sensor. The S2 data products were downloaded from the Sentinel
Open Access Hub platform (ESA, 2019) in the UTM zone 43N projection coordinate system as a Level 1C tile (100 km × 100
km). The Level 1C product level consists of ortho-images radiometrically corrected top-of-atmosphere reflectance values, and
geometrically corrected based on a refined geometric model. We used the red band (band 4) of S2, because it is appropriate for
glacier monitoring (Kääb et al., 2016) and used as a reference for band-to-band co-registration (Table 1; Gascon et al., 2017).

Landsat 8 utilizes the Operational Land Imager (OLI) sensor, with a swath width of 185 km (Table 1), and has a 16-day
temporal resolution. We downloaded L8 images from the United States Geological Survey's Earth Explorer platform (U.S.
Geological Survey, 2019) as L8 L1T product. The processing of L8 products is similar to that of S2 data with respect to
radiometric and geometric corrections, ortho-rectification and re-sampling to a map grid. We chose the L8 panchromatic band
(band 8), due to its 15m spatial resolution, (Table 1). All available optical images where used to map the evolution of the
paraglacial lake and terminus advance.

We used the Shuttle Radar Topography Model (SRTM,  U.S. Geological Survey, 2014) DEM in combination with three
DEMs, one in 2017 and two in 2019, that were calculated with Planetscope, GeoEye-1 and WorldView-2 images as presented
by Aati and Avouac (2020). To constrain the bedrock topography of Shisper glacier, we used the three different bed elevation
models proposed in Farinotti et al. (2019) and averaged the three results, as suggested by Farinotti et al. (2017).

## 4   Data processing

The data processing consists of three main steps (processing chain in Fig. 3) which we detail in the following subsections and in
the supplement: (A) the data is prepared by clipping the images to the study area, and removing the cloudy images; (B) surface
displacement maps are calculated from correlating consecutive images with the COSI-Corr software package (Leprince et al.,
2007). For each image pair there are three output, East-West and North-South displacement maps and a map of signal-to-noise





**Table 1.** Overview of satellite data products analyzed in this study.

| Satellite | Landsat-8 | Sentinel-2A/2B |
|---|---|---|
| Sensor | OLI | MSI |
| Processed data level (Product level) | USGS level L1T | ESA level 1C |
| Spectral band | Panchromatic (B8) | Red band (B4) |
| Wavelength $\lambda$ ($\mu$ m) | 0.50 to 0.68 | 0.65 to 0.68 |
| Sampling resolution (m) | 15 | 10 |
| Orbit revisit (days) | 16 | 10 (5 staggered) |
| MSI tile or Path-Row | 149-35 | T43SDA |
| Time period | 05/05/2013-04/01/2019 | 05/21/2016 -09/10/2019 |
| Number of images (displacement maps) | 52 (51) | 48 (47) |

ratio (SNR); (C) The data is filtered and artifacts removed, for example systematic offset between sensors. The processing chain was designed to allow for a nearly automated processing of all the available data (L8 and S2).

Until the post-processing (step C, Fig. 3), images are processed independently for either satellite. The correlation of S2
and L8 images are not included due to systematic othorectification artifacts. The USGS and ESA only release ortho-images and we would need access to non-rectified images in order to circumvent these artifacts. We only use pairs with the smallest consecutive time span for the correlation, which leads to a sequence of $n-1$ displacement maps for each satellite with $n$ different acquisition dates. This strategy was chosen so as to get a complete times series for a minimal computational volume.

### 4.1 Image pairs correlation

This study implements the COSI-Corr frequency correlator (Leprince et al., 2007), which is better adapted to measure small displacement over short time spans than the statistical correlators included in other image correlation packages (e.g. MicMac, CIAS, Medicis; see Heid and Kääb (2012); Rosu et al. (2015)). This correlator works in two processing steps. (1) The first step provides a coarse estimate using a large sliding window. (2) A subpixel accuracy is obtained from a second correlation using a smaller window. Three parameters are chosen, the step which determines the spatial resolution of the displacement map and
the sizes of the first and second sliding windows. The values used in this study are listed in Table 2. With these parameters we obtain independent measurements of surface velocities with a ground resolution of about half the smaller window size, i.e. 160 m for S2 and 240 m for L8. Our velocity maps, which have a pixel size of 60m, are thus oversampled by a factor 3 (S2) or 4 (L8). The image correlation process yields two displacement maps, each representing one of the horizontal displacement (East-West and North-South), as well as a measure of the correlation quality (e.g. Signal-to-Noise Ratio, SNR). The proposed work
flow does not depend on the choice of a particular image correlation algorithm and be based on any other image correlation software.



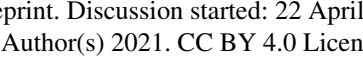

**Figure 3.** Work flow for image processing, displacement map preparation, and velocity map post-processing.





**Table 2.** Summary of the parameters used for image correlation and displacement filtering.

| Parameter | | Value |
|---|---|---|
| Initial window size | | 64 x 64 pix |
| Final window size | | 32 x 32 pix |
| Step | S2 – data | 6 x 6 pix (final GSD 60 m) |
| | L8 – data | 4 x 4 pix (final GSD 60 m) |
| Iteration | | 4 |
| Mask threshold ($T_{\mathrm{snr}}$) | | 0.9 |
| Resampling | | True |
| Grid | | True |
| Window size $w$ | | 21 |
| $s$ | | 2 |
| $T_{\mathrm{Med}}$ | | 1.5 |

## 4.2 Filtering and registration improvement

After the image pair correlation, three post-processing steps take place: (1) spatial data filtering to remove outliers from each displacement map and fill in missing data, (2) offset correction, to reduce systematic offsets due to incorrect ortho-rectification,

and (3) reduction of noise level using a principal component analysis (PCA). This noise reduction takes advantage of the correlations in space of the displacement field.

To perform the spatial data filtering and data filling, we designed a specific multi-scale spatial filtering process to remove erroneous measurements due to clouds, snow cover or surface water. This post-correlation tool, entitled Local Multiscale Filter (to be included in forthcoming version of COSI-Corr), consists of 4 steps (Appendix A):

• **Step 1** : Filtering outliers with high uncertainty. We exclude measurements with low SNR either by removing measurements with (1) a SNR value smaller than a certain threshold $T_{\mathrm{snr}}$ or (2) outliers beyond a user-selected multiple of the standard deviation ($T_\sigma$).

• **Step 2** : Weighting of each displacement map with SNR.

• **Step 3** : Filtering outlier displacement measurements. For each displacement measurement of the correlation map in position

$(x, y)$, we define a neighborhood window $w(x, y, s)$, where the $s$ parameter is user-defined and controls the size of the window. Then, the algorithm removes values larger than the $T_{\mathrm{Med}} \times \mathrm{Med}$, where $\mathrm{Med}$ is the median value of the displacement measurement in $w$ and $T_{\mathrm{Med}}$ is a user-defined threshold, taken as $T_{\mathrm{Med}} = 1.5$ here. To avoid filtering out reliable local values, a validation process is applied for each $w$, where the window is only considered valid if at least 70% of the values lie within the confidence interval defined by $\pm |T_{\mathrm{Med}} \times \mathrm{Med}|$, where $|\cdot|$ denotes the absolute value.

Otherwise, the window scale changes depending on the scale factor setting $f$.





- **Step 4** : Filling in missing data. We estimate the missing values created during the steps above. The algorithm flags filtered values as missing data (e.g., NaNs) in defining an uncertainty matrix. For each window, NaN values are initiated by the $\mathrm{Med}$ value of window w and the weight of this measurement is defined as the mean of SNR values in the neighboring window w. Otherwise the original displacement value weight remains consistent with the original SNR value. This step relies on the assumption of local spatial coherence across neighboring windows w, thus that the displacement changes gradually rather than abruptly.

After the spatial filtering, we apply a correction for the offsets caused the by the mis-registration of available data products. These errors can be mitigated by optimizing the co-registration of the images during the orthorectification procedure (e.g. Avouac and Leprince, 2015). However, accounting for mis-registration is not possible with S2 and L8 data, and we simply apply a linear correction to the correlation maps which provides a first-order systematic correction.

The final step consists in filtering the data. This step relies on the assumption that spatial pattern of surface velocity is highly correlated. Spatial ice flow variations occur at scale much larger than the ground sampling distance of our velocity maps. To identify these correlations, we use a PCA and reconstruct the time series with the limited number of components needed to recover the data variance. We retain the $k$ first components needed to account for 90% of the variance. We convert the displacement maps to surface velocities by dividing the elapsed time between the paired images.

Velocity maps are then stacked on the same data cube $\boldsymbol{D}$ according to the target image date, data cube that is decomposed in principal components. This procedure does not involve any smoothing of the temporal variations of ice velocities. Abrupt variations affecting a limited area are however filtered out because they would not contribute much to the data variance.

## 5  Results

### 5.1  Result overview

We present the surface velocities of the two glaciers together to allow for a relative comparison (Fig. 4). Mochowar glacier only show relatively regular seasonal variations (see Figs. S3 and S4). Shisper glacier shows much more variability and we can distinguish various patterns: (1) relatively slow velocities during the quiescence and between speed-ups (Fig. 4a), (2) spring speed-ups (Fig. 4b, d and e; see also Abe and Furuya, 2015) (3) fall speed-up (Fig. 4c), (4) surge onset (Fig. 4f) and (5) different phases of the surge (Fig. 4g,h,i and j). Spring speed-ups are seen clearly in both glaciers, but the other forms of velocity patterns are unique to Shisper glacier.

During the quiescence, velocities are generally comprised between $\sim 0.35\,\mathrm{m\,d^{-1}}$ and $\sim 0.8\,\mathrm{m\,d^{-1}}$ in the upper and main trunks. Surface velocities during the spring speed-ups increase in amplitude from $\sim 1\,\mathrm{m\,d^{-1}}$ in the main trunk in 2015 (Fig. 4b), to $\sim 2\,\mathrm{m\,d^{-1}}$ in spring 2016 (Fig. 4d), and to $\sim 3\,\mathrm{m\,d^{-1}}$ in spring 2017 (Fig. 4), eventually leading up to the surge onset by the end of 2017 (Fig. 4f). As the surge starts, velocities are consistently greater than $\sim 1\,\mathrm{m\,d^{-1}}$ in the lower part of the upper trunk and below. Before the surge onset, we can distinguish seasonal speed-ups in the fall (Fig. 4c), showing velocities greater than





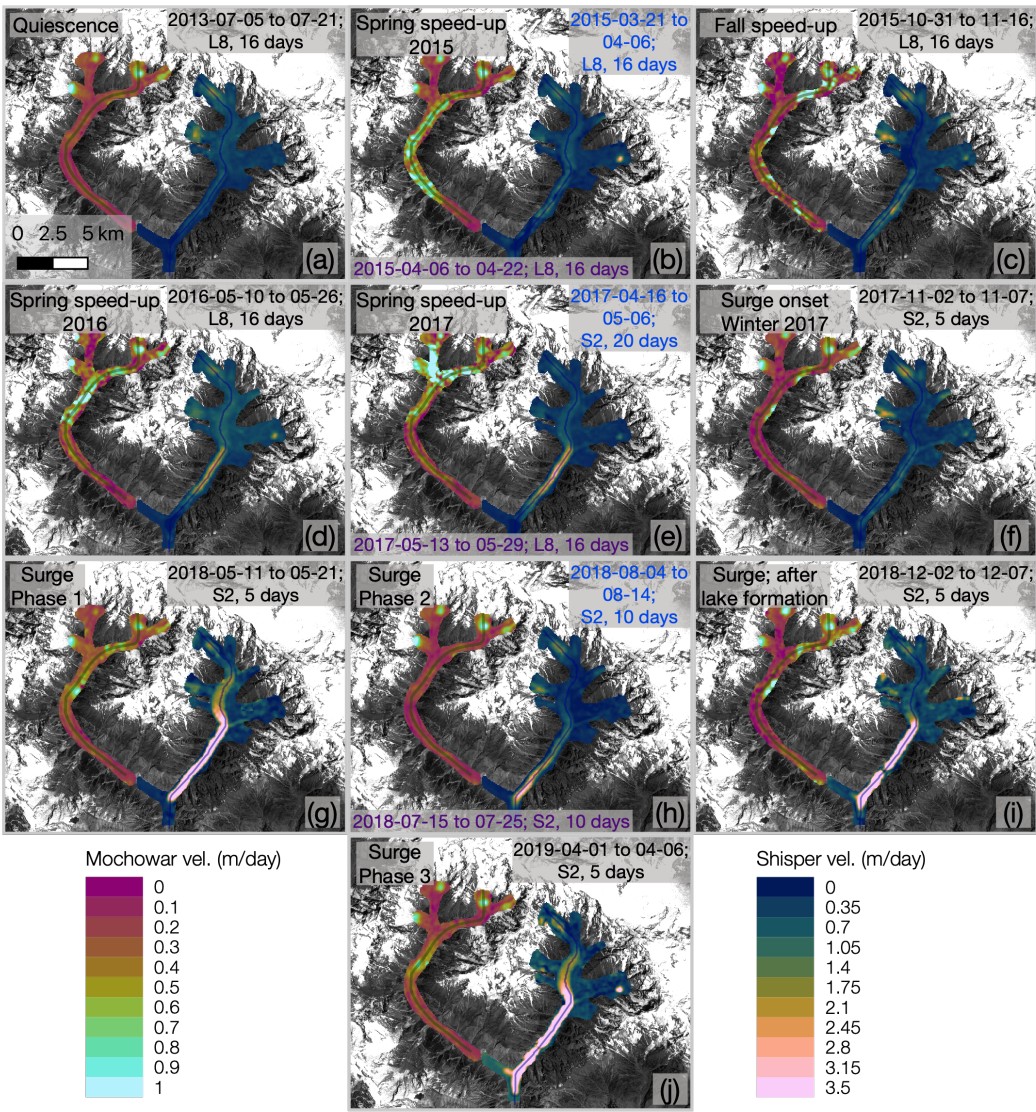

**Figure 4.** Velocity maps showing the dynamic behavior of Mochowar and Shisper glaciers between 2013 and 2019. (a) Quiescence in the absence of seasonal speed-up; (b) Spring speed-up; (c) Fall speed-up; (d) Enhanced velocities during spring speed-up in 2016; (e) Enhanced velocities during spring speed-up in 2017; (f) Slow fall onset of the surge; (g) Phase 1 of the surge; (h) Phase 2 of the surge; (i) Phase 2 of the surge after lake formation; (j) Phase 3 of the surge. The choice of velocity snapshot is aimed at displaying the most notable velocity patterns of Shisper glacier whereas Mochowar is used as a reference. Note that the velocity maps for each glaciers are not always taken over the exact same time period in order to display the cleanest velocity maps In which case, the time stamp is shown in purple for Mochowar and Blue for Shisper. The colorbars have different scales for either glacier to better display velocity patterns. Perceptually uniform color maps are used in this figure to prevent visual distortion of the data (Crameri, 2018a, b).



$\sim 1\,\mathrm{m\,d^{-1}}$ throughout the main trunk. The timing of these fall speed-ups correlates well with that of the onset of the surge in 2017 (Fig. 4f).

Phase 1 of the surge displays velocities greater than $1\,\mathrm{m\,d^{-1}}$ over most of the glacier with the entire main trunk flowing faster than $6\,\mathrm{m\,d^{-1}}$ (Fig. 4g). Phase 2 is characterized by significantly slower velocities, yet the main trunk keeps on flowing at least at $1\,\mathrm{m\,d^{-1}}$ with a maximum above $3\,\mathrm{m\,d^{-1}}$ towards the terminus (Fig. 4h). An interesting event during phase 2 is the formation of a paraglacial lake once Shisper glacier blocks the valley from Mochowar (mid-November 2018). The lake filling is associated with a short-lived increase in surface velocities ($\geq 6\,\mathrm{m\,d^{-1}}$) in the lower main trunk and terminus area (Fig. 4i). Finally, phase 3 shows very similar velocity patterns to phase 1 (Fig. 4j), with most of the glacier flowing faster than $1\,\mathrm{m\,d^{-1}}$, and the main trunk flowing faster than $7\,\mathrm{m\,d^{-1}}$. In addition, terminus velocities are greater than $3\,\mathrm{m\,d^{-1}}$ as the surge bulge progresses.

## 5.2 Shisper glacier surge and its build-up

The surface velocities of Shisper glacier exhibit a marked seasonal pattern from the beginning of the timeseries and throughout the surge (Fig. 5a). This seasonal signal prior to the surge consists of surface velocities exceeding $1\,\mathrm{m\,d^{-1}}$ for a significant portion of the glacier profile, compared with velocities remaining between $\sim 0.35\,\mathrm{m\,d^{-1}}$ and $\sim 0.8\,\mathrm{m\,d^{-1}}$ otherwise. This seasonality is expressed, biannually, between March and June, i.e. a spring speed-up, and another velocity increase between late September and November, that we will thereafter call fall speed-up. It appears that the presence of snow introduces a significant amount of noise in the data between fall and winter, especially at high elevations, i.e. between kilometer 0 and 2 along the profile. The fall speed-up is clear in 2013 and 2015, faint but present in 2016, and our results are inconclusive for 2014.

In the absence of field measurements, we use the temperature reanalysis data from the Modern-Era Retrospective Analysis for Research and Applications, version 2 (MERRA-2; Gelaro et al., 2017) to confirm that this seasonality is linked with glacier surface melt, and thus hydrology (Fig. 5c). The spring speed-ups occur soon after the daily-averaged two-meters-above-ground temperature is estimated to exceed $0°\,\mathrm{C}$ at 4176m a.s.l. (mean altitude of reanalysis grid cell) in the basin. The main trunk is located at least one kilometer below this elevation, which suggests that it is undergoing significant surface melt. Similarly, the fall speed-ups occur soon after the reanalysis temperature at 4176m. a.s.l. drops below freezing, hence when water supply from the glacier surface is expected to be shutting off and the drainage system of the glacier shutting down.

Defining a surge based on surface velocities is somewhat arbitrary. A criterion that is pervasive in the literature is that surface velocities increase by an order of magnitude during the surge (e.g. Clarke et al., 1986), compared to that during the quiescence. Thus, we normalize surface velocities by the average of velocity during the quiescence, i.e. from the beginning of the timeseries until the slow onset of the surge in November 2017 (Fig. 5b). We consider that when normalized velocities reach or exceed 10, surge-level velocities are reached. This normalization further highlights the gradual increase in the amplitude of the spring speed-up as the surge becomes more imminent. In 2016 the speed-up shows velocities that are five to six fold the average quiescence velocities and in 2017 the ten-fold threshold is reached (Fig. 5b). Aside from the 2016 and 2017 spring speed-ups,

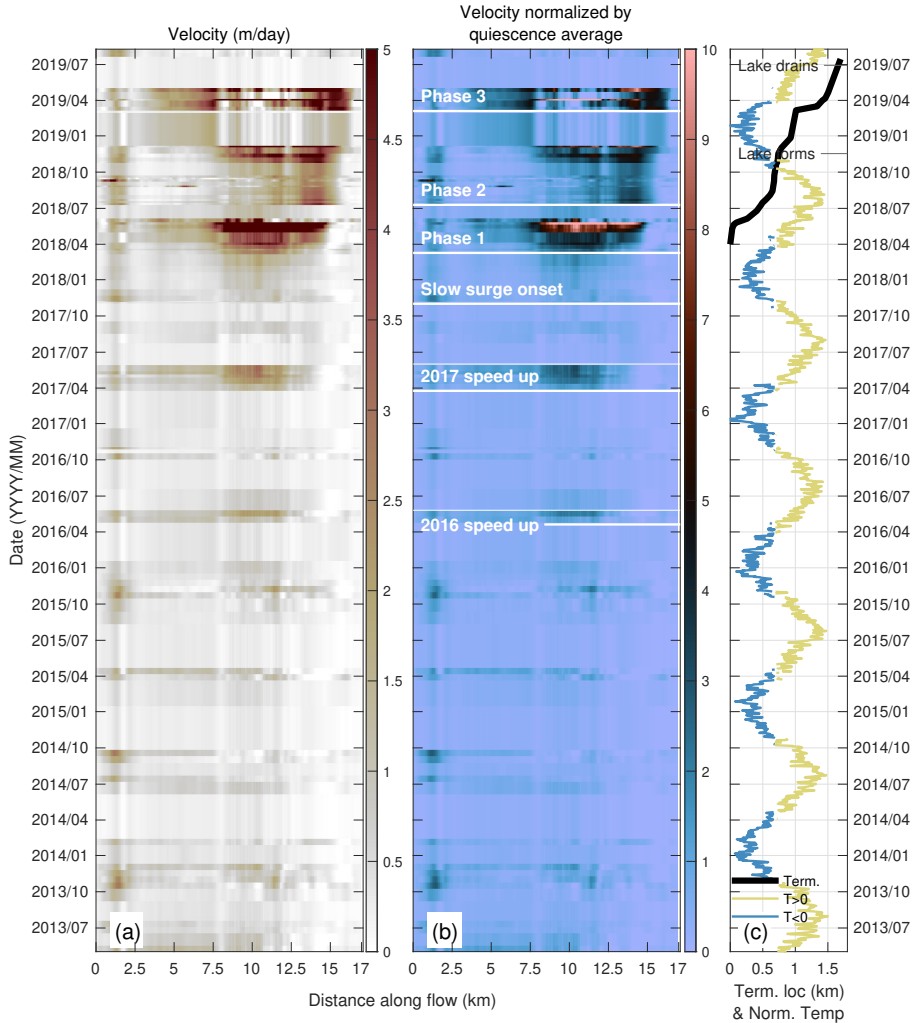

**Figure 5.** Time series of (a) surface velocity and (b) normalized surface velocity along the flowline of Shisper glacier (Fig. 2) from 2013 to 2019. The plots were produced from the data cube of L8 and S2 images with the PCA reconstruction. (c) Time series of terminus position advance and MERRA-2 satellite reanalysis of maximum daily temperature data for the study area at an equivalent elevation of 4176 m a.s.l.; temperature are normalized by the maximum value, and blue and yellow represents temperature below and above $0°$, respectively. In (b) velocities are normalized with respect to average surface velocity during the quiescence $\overline{u}_{\mathrm{quiesc}}$. Normalized velocities exceeding 10 are considered surge velocities. The major surface velocity patterns that we identified are shown with the white labelled lines in (b). Perceptually uniform color maps are used in this figure to prevent visual distortion of the data (Crameri, 2018a, b).

surface velocities seldom exceed $\sim 2\,\mathrm{m\,d^{-1}}$ before the surge onset, which is similar to surface velocities at Mochowar glacier. Note that the highest velocities are located in the main trunk region of the glacier.





In November of 2017, velocities increase significantly in the main trunk (Fig. 5a) reaching levels of four to five-fold the average quiescence velocities (Fig. 5b). While the surface velocities in the main trunk in the winter of 2017-2018 remain $\sim 2\,\mathrm{m\,d^{-1}}$, they are significantly larger than previous winter velocities in our record. The first significant surge phase (Phase

1) starts in the main trunk in late March 2018 with velocities at least an order of magnitude larger than quiescence average (Fig. 5b), peaking at $\sim 13\,\mathrm{m\,d^{-1}}$ in late May. As Phase 1 progresses, the surge velocities also gain the upper trunk and possibly the ice fall by June 2018. Interestingly the Phase 1 of the surge coincides with the timing of spring speed-ups at Shisper glacier in previous years (Fig. 5c). During this period, however, little terminus advance is observed because the surge bulge has merely reached the terminus. Terminus advance starts in June 2018 which is unfortunately a period during which data quality is poor.

At the end of June 2018 Shisper advances into the adjacent valley.

Phase 2 starts at the end of the 2018 summer and is characterized by surface velocities slower than in Phase 1. Surface velocities generally decrease until the late fall 2018, but remain at surge levels in the lower part of the main trunk and the terminus (downstream from kilometer 11). The slow down is most noticeable between kilometers 7.5 and 11. The terminus advances by 0.5 km over these six months. A notable event is the increase in surface velocities (November and December of

2018) in the main trunk and terminus soon after the formation of the lake in the adjacent valley (Fig. 5c). This glacier-wide velocity increase is also marked by a terminus advance of a few hundred meters.

Phase 3 exhibits the most dramatic terminus advance ($\sim 500$m in less than a month) which occurs in the spring of 2019 and is associated with particularly high surface velocities. Surge-level velocities reach the upper trunk and the ice fall, and maximum velocities reach $\sim 12\,\mathrm{m\,d^{-1}}$ (early May 2019) which is slightly slower than during Phase 1 ($\sim 13\,\mathrm{m\,d^{-1}}$ in June 2018). Phase

3 results in Shisper terminus advancing $\sim 0.7$ km in 3.5 months. We note that artifacts from cloud cover or shadows may be responsible for the low velocities areas calculated between km 7.5 and 12.5 over that period. Similarly, the low velocities observed between May and July 2019 are likely the result of poor image quality or correlation.

The paraglacial lake in the Mochowar valley is located $\sim 2.5$ km from the terminus (Fig. 2) and fills consistently from its formation in November 2018 until its drainage on June 23th 2019 (Pamir Times, 2019). The lake drainage appears to coincide

with the termination of the surge. In the second half of July and early August (last images used), the only parts of the glacier that maintain velocities higher than the quiescence average, are in the ice fall or the terminus regions. This is likely because these sections of the glacier are still adjusting to the post surge ice topography, as described in the next section.

### 5.3 Elevation changes between 2000, 2017 and 2019

Both Shisper and Mochowar glaciers show increasing surface elevations between 2000 and 2017, at the exception of the area of their terminus constituted of ice left over by previous surges (Fig. 6a, b, and c). This increase in surface elevation can be likened to a positive mass balance, and is consistent with the Karakoram climatic anomaly (e.g. Gardelle et al., 2012; Kääb et al., 2015). The height increases on both glaciers ($\sim 10$ to 50m over 17 years, Figure 6a, b, and c), are consistent with those measured by Gardelle et al. (2012, Fig. 3 therein), that are on the order of several tens of meters in about a decade. It
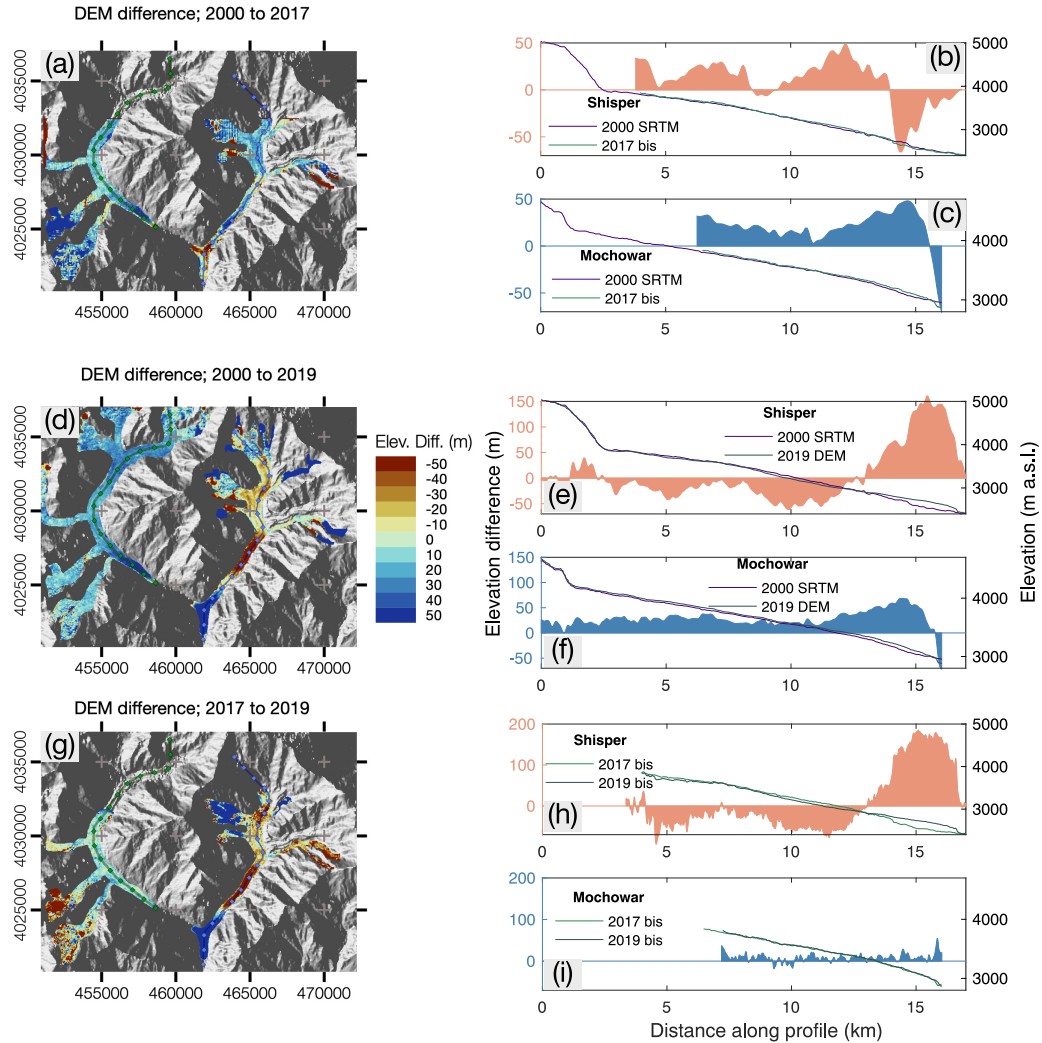

**Figure 6.** Surface elevation change at Shisper and Mochowar between 2000, 2017 and 2019. The left column are map views of the DEMs of differences and the right column displays the elevation differences and surface elevation along the profiles. The elevation difference is shaded above and below 0 to highlight mass gains (above) and mass loss (below). (a), (b) and (c) Show the difference between the SRTM DEM of 2000 and the DEM created for 2017 with PlanetScope images (2017 bis); (d), (e) and (f) the difference between the SRTM DEM of 2000 and the DEM extracted from GeoEye-1 and WorldView-2 Stereo pair imagery in 2019 (2019 DEM); (g), (h) and (i) the difference between the DEM created for 2017 (2017 bis) and 2019 (2019 bis) with PlanetScope images. Note that for the 2017 and 2019 PlanetScope DEMs the available imagery did not allow to create DEMs covering the entirety of the glaciers, hence the data gap in the upper reaches of either glacier. The creation of the DEMs is detailed in Aati and Avouac (2020). Perceptually uniform color maps are used in this figure to prevent visual distortion of the data (Crameri, 2018a, b).





is important to note, however, that the present study area is located on the western edge of the Karakoram, region generally expected to undergo slightly negative mass balance budgets (Gardelle et al., 2012; Farinotti et al., 2020).

Because the latest surge of Shisper glacier occurred between 2000 and 2005 (see S2), the 2000 SRTM DEM is likely a representation of the surface of the glacier towards the end of its quiescence phase or beginning of the surge. The DEM computed based on 2017 images does not cover the ice fall, unfortunately, but it shows a gain in elevation across the upper
and main trunk. By contrast, the terminus shows a clear mass loss between 2000 and 2017 (Fig. 6b) where the stagnant ice from previous surges is expected. The area where the most elevation gain takes place is between kilometer 10 and 13, which corresponds to the lower part of the area where velocities are the fastest. The elevation gain in this area could also be the result of mass redistribution from the surge in the early 2000's that was smaller than the one described here.

Since 2000 almost the entirety of the glacier surface has lowered as a result of the surge (Fig. 6d-i), at the exception of the
terminus and lower section of the main trunk that acted as the receiving area during the surge (Fig. 6e and h). The area that undergoes the most lowering is part of the main trunk, between km 8 and 13, located just upstream from where the observed velocities are the fastest (Fig. 6e and h). The surge emplaced up to 200m of ice in the terminus region, with at least 100m over the last 4km of the glacier (Fig. 6e and h) which explains the dramatic advance and the fact that the terminus seems to keep on advancing even after the surge has terminated.

### 5.4 Mass balance of Mochowar glacier

Similarly to Shisper, the elevation change at Mochowar points to a positive mass balance since 2000, also consistent with the Karakoram anomaly (e.g. Gardelle et al., 2012; Kääb et al., 2015). The fact that both glaciers show similar patterns point to a climatic driver for surface elevation change at both glaciers, rather than to an individual dynamic behavior (e.g. Roe and O'Neal, 2009). The drop in elevation seen in the 0.5km of the profile is associated with the melting of stagnant ice left by a
prior surge (see S2). It is worth noting that Mochowar and Shisper were connected until 2005. However, the last documented surge of Mochowar glacier prior to the current study dates back to the 1970's (Bhambri et al., 2017).

The fact that the ice thickness increases by several tens of meters within $\sim 5\,\mathrm{km}$ of the terminus suggest that a slow surge actually took place (see Section 5.3), otherwise the thickening would be expected to be the largest at higher elevations. In addition, Mochowar glacier shows significant seasonal signal with spring speed-up every year (Fig. S3 and S4). The data is
noisy, but there seem to be a slight increase in velocities in 2018 (Fig. S3).

## 6 Discussion

### 6.1 Spatio-temporal evolution of surge dynamics

The surge of Shisper glacier shows a dramatic terminus advance. The terminus advanced by over 1.7 kilometers, emplacing up to 200 meters of ice in the terminus region and over 100 meters over 4km (Figs. 5c and 6). In comparison, the Haut-Glacier
d'Arolla, Switzerland, was 180 m at its thickest and 4 km long in the 1990's surge (Sharp et al., 1993). Interestingly, Shisper





surged in early 2000's, but this older surge was much less pronounced as the terminus stopped $\sim 700\,\mathrm{m}$ up-valley from the surge presented here (see S2). We do not have enough information to discuss the differences between the two surges, but it suggest that the fast flow part of a surge cycle can vary over time. The possible difference between surges of a single glacier could challenge the idea that glaciers in different climatic regions have different surging mechanisms (e.g. Murray et al., 2003;

Quincey et al., 2015), however suggest that a single mechanism has different manifestations.

The velocity timeseries from Shisper glacier suggests a clear correlation between surge dynamics and glacial hydrology. We find that the dynamic evolution of the glacier can be decomposed in the following patterns (Figs. 5 and 7): (1) A regular seasonal signal both in the spring and the fall until spring 2016, (2) a gradual increase in spring speed-ups magnitude in 2016 and 2017, (3) slow onset of the surge in November 2017, (4) Phase 1 of the main surge coinciding with the 2018 spring

speed-up, (5) Phase 2 shows a slow down in the surge with a short-term acceleration when a significant lake forms, (6) Phase 3 shows a significant increase in velocities that coincides with the 2019 spring speed-up, and (7) the surge terminates at the end of June 2019 in unison with the lake drainage. The consistent correlation between surge events and expected changes in glacial hydrology suggests that the surge represents a dynamic state where basal sliding is enhanced, but remains sensitive to hydraulic forcing.

Velocity fluctuations during surges appear to be commonplace when observations with a high-enough temporal resolution are available. The most striking example is the surge of Variegated glacier in 1982–1983, where Kamb et al. (1985) recorded velocity fluctuations both at the seasonal scale, and within hours. They furthermore identify a clear link between glacier hydrology and velocity fluctuations during the surge, while they refer to these events as mini-surges. A two-phase surge with peak velocity associated with surface melt-water production was also identifed for Kyagar glacier, Chinese Karakoram (Round

et al., 2017). A gradual increase in surface velocities in the several years leading up to the surge of Austfonna/Basin-3, Svalbard, in the early 2010s, is reported in Dunse et al. (2015). Again, velocity peaks akin to spring speed-ups are present, even during the surge. However, the amplitude of the speed-ups does not appear to change significantly (see Fig. 2 in Dunse et al., 2015). Another example of seasonal velocity changes during a surge comes from Hagen Bræ, North Greenland (Solgaard et al., 2020), where both summer and winter speed-ups are also documented.

An interesting outcome of the current study is that using only L8 or S2 would have led to an incomplete picture of the dynamics of Shisper glacier. The L8 data holds the information about the dynamics prior to the surge and the S2 data shows the intricacies of the surge onset and its different phases (Fig. S5). The main reason why we are able to identify the different dynamic events leading up to and during the surge is the quality and the spatio-temporal resolution of the velocity timeseries. Based on the differences between L8 and S2 derived datasets (Fig. S5; see also Bhambri et al. (2020) Figure 3), we suggest

that the conundrums related to specific surge mechanisms and evolution in different regions of the world (e.g. Murray et al., 2003; Quincey et al., 2015) are in part due to observational limitations. It is also important to note that even in our dataset such limitations exist as velocity changes can be significant at an hourly timescale (e.g. Kamb et al., 1985). The data quality, furthermore allows us to identify the main trunk as the location of the surge initiation and to suggest that, at Shisper, the dynamic response of the rest of the glacier is driven by the dynamics of the main trunk.





## 6.2 Surge trigger

Two main surge trigger mechanisms have been proposed: thermal or hydraulic (e.g. Meier and Post, 1969; Clarke et al., 1984; Jiskoot et al., 2000; Murray et al., 2000, 2003; Sevestre et al., 2015). The thermal trigger consists in an event that thaws a glacier bed that was previously frozen enabling widespread sliding to take place (e.g. Clarke et al., 1984; Murray et al., 2000). The hydraulic trigger represents an event that leads to increased basal water pressure reducing bed resistance and enabling fast sliding to take place (e.g. Kamb et al., 1985; Murray et al., 2003; Sevestre et al., 2015). While it remains a broadly recognized hypothesis (e.g. Farinotti et al., 2020), there is little direct evidence to support the thermal trigger of cyclic surges (Clarke et al., 1984; Frappé and Clarke, 2007; Sevestre et al., 2015). A slow build-up and termination (e.g. Murray et al., 2003) is a widely used characteristic to determine the thermal trigger (e.g. Quincey et al., 2015). On the other hand, the hydraulic trigger is expected to produce more sudden behavior.

Our evidence suggests that the surge trigger for Shisper is controlled by glacier hydrology and is similar to that of the 1982–1983 surge of Variegated glacier. The initial surge onset appears in the Main trunk in fall (Fig. 5), and correlates with fall speed-ups observed in previous years (Figs. 5 and 7). This corroborates the idea put forward by Kamb et al. (1985) that the closure of an established efficient drainage system leads to the re-pressurization of the ice–bed interface and an increase in velocities (Fig. 7b). Then surge velocities are only reached in the spring of 2018, again highlighting the link with hydraulic processes and the typical timing of spring speed-ups. Finally, the thermal trigger can confidently be ruled out for Shisper as the presence of both a spring and fall speed-ups in the several years leading up to the surge suggests that sliding occurs at the bed.

## 6.3 Interplay between lake drainage and surge termination

The surge termination coincides with the drainage of the glacier-dammed lake on June 23, 2019 (Figure 5c; Pamir Times, 2019; Rashid et al., 2019). Cloud cover hampered our ability to utilize images collected between these dates. Several coincident processes occurs, which indicate that either the lake drainage affects the surge termination or vice-versa. Surge termination is notoriously associated with a flood, whether it results from a glacier dammed lake or subglacially stored water (e.g. Kamb et al., 1985; Round et al., 2017; Steiner et al., 2018; Zhan, 2019). During a surge a distributed drainage system is necessary to maintain high water pressures. As a channelized drainage system forms, it is able to drain the water out of the glacier bed more efficiently, decreasing overall water pressures (e.g. Röthlisberger, 1972; Iken and Bindschadler, 1986; Björnsson, 1998; Mair et al., 2002; Werder and Funk, 2009), consequently terminating the surge (e.g. Kamb et al., 1985). In the case of Shisper, two scenarios are possible to explain this simultaneity: (1) the lake drainage initiated the onset of a channelized drainage under the whole glacier, or (2) the drainage of the water stored by the glacier during the surge opened waterways for the lake to drain. In the present study, we lack data to establish a causality between the two phenomenon.

Here and in other studies (Round et al., 2017; Steiner et al., 2018) lakes lay only about $1$–$3\,\mathrm{km}$ from the glacier terminus. The drainage of these lakes is synchronous with the termination of the surge. The lake drainage on June 23th, 2019 only mildly affected Hassanabad Village, but severely damaged the Karakoram Highway (Pamir Times, 2019). The lake formed and drained again in 2020, this time a month earlier, around May 29th, which could create a recurring hazard. Determining the



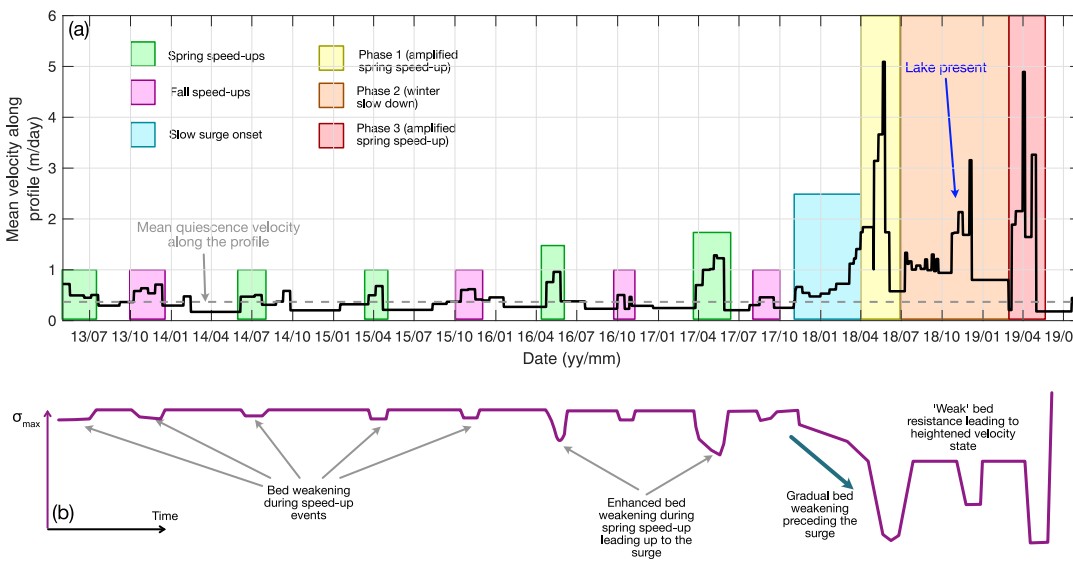

**Figure 7.** (a) Spatial average of surface velocities along the profile of Shisper glacier as a function of time and (b) conceptual representation of the maximum resistive stress $\sigma_{\mathrm{max}}$.

**Table 3.** Parameters used to fit the generalized sliding relationship and comparison with values published in the literature. We use the symbol - when no values are specified.

| Relationship | $u_{\mathrm{t}}$ | $\sigma_{\mathrm{max}}$ | $p$ | $q$ |
|---|---|---|---|---|
| Main glacier low bound $f_a$ | $3\,\mathrm{m\,day}^{-1}$ | $80\,\mathrm{kPa}$ | 2 | 1 |
| Main glacier high bound $f_b$ | $0.4\,\mathrm{m\,day}^{-1}$ | $500\,\mathrm{kPa}$ | 5 | 3 |
| Gagliardini et al. (2007) | - | - | 1–4 | 1–3 ($\geq 1$) |
| Zoet and Iverson (2020) | $\sim 0.1\mathrm{m\,day}^{-1}$ | $\sim 86 - 100\,\mathrm{kPa}$ | 5 | 1 |
| Minchew et al. (2016) | - | $100 - 150\,\mathrm{kPa}$ | - | 1 (i.e. plasic bed) |

causality between the lake drainage and surge-terminating flood would be important to determine the size of expected floods. If the volume of the water stored is added to that of the lake, the hazard could be much more important that if only the lake is considered.


## 6.4 Towards a unified glacier sliding relationship

We present a first-order assessment of a generalized sliding relationship (Eq. 6) pending some simplifying assumptions. We use the DEMs available and the models of bed topography (Farinotti et al., 2019) to estimate driving stress ($\tau_{\mathrm{d}} = \rho g h_{\mathrm{ice}} \nabla S_{\mathrm{ice}}(x,y)$, with $\rho$ the ice density, $g$ the gravitational acceleration, $h_{\mathrm{ice}}$ the ice thickness and $\nabla S_{\mathrm{ice}}(x,y)$ the ice surface gradient) that can



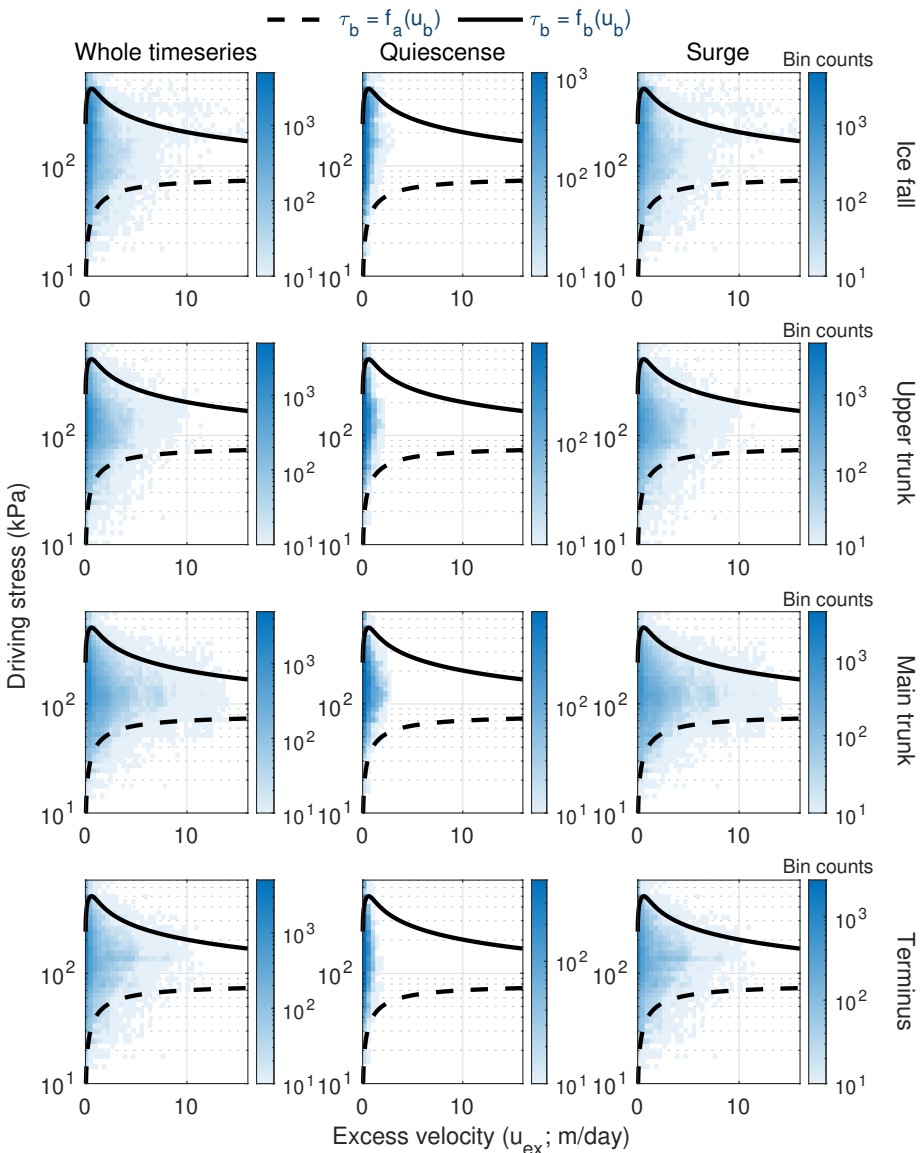

**Figure 8.** Relationship between driving stress, used as a proxy for shear stress, and excess velocities, used as a proxy for sliding velocities. The data on each axis is separated in 40 bins and the plot color represents the number of data points within a bin. Possible bounds for a generalized sliding relationship (Eq. 6) are shown in the full and dotted lines and the associated parameters are listed in Table 3. Note that the Y-axis is logarithmic and the X-axis is linear and the effect of axes scaling on data visualization is shown in Figs. S8 and S9.

be used to approximate basal shear stress (Cuffey and Paterson, 2010). To ensure that the velocity change used in the slip relationship is associated with sliding, and that deformation velocity only account for a negligible fraction of the signal, we

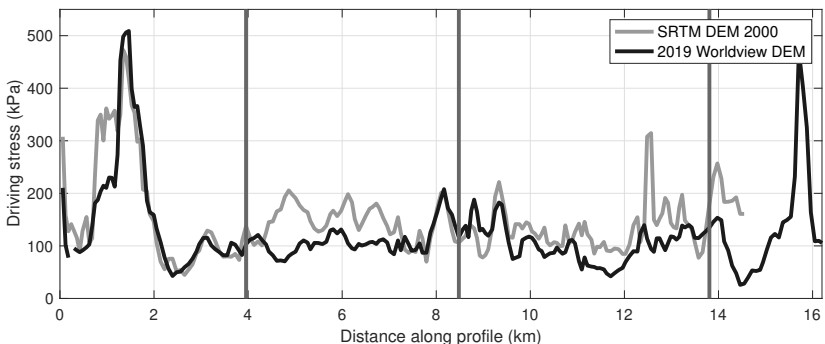

**Figure 9.** Driving stress profiles as a function of DEMs before (SRTM) and during (Worldview 2019) the surge. Vertical lines show the different section of the glacier, from left to right, the ice fall, the upper trunk, the main trunk, and the terminus (see Fig. 2).

use the difference between surface velocities and the mean quiescence velocities ($u_{\mathrm{ex}} = u_{\mathrm{surf}} - \overline{u}_{\mathrm{quiesc}}$). We term this velocity difference the excess velocity ($u_{\mathrm{ex}}$). The mean quiescence velocity ($\overline{u}_{\mathrm{quiesc}}$) is calculated by time averaging the entire velocity field for each velocity map until the surge onset in November 2017. We then plot an estimation of the relationship $\tau_{\mathrm{b}}$ versus

$u_{\mathrm{ex}}$ for our dataset (Fig. 8). Equation 6 enables us to identify 4 free parameters $u_{\mathrm{t}}$, $\sigma_{\mathrm{max}}$, $p$ and $q$ that we can tune to bracket the data.

The main source of uncertainty here is the ice geometry, we have no field data for bed elevation and we lack data on the evolution of the ice surface during the surge. We choose to use the 2000 SRTM DEM to calculate driving stresses from the beginning of the timeseries until the lake forms. Afterwards, we use the 2019 DEM. This is a somewhat arbitrary change based

on the expected geometry evolution. It is important to note that using either DEM for the driving stress approximation does not alter the results significantly (Figs. S6 and S7).

We divide the velocity dataset into four spatial zones, as delimited in Fig. 2 and Fig. 9, and two periods, the quiescence and the surge (see Fig. 8). The ice fall and the front of the terminus show particularly high driving stresses because of surface slopes greater than $30°$(Fig. 9). We find that the relationship between driving stress and velocity is consistent across the different

glacier regions over the span of our data, and clearly shows a ceiling in driving stresses at high excess velocities (Fig. 8).

The ice fall region shows particularly high driving stresses and excess velocities are more scattered than for the rest of the dataset. The presence of clouds and snow over topography at high elevations weakens image correlation, a likely source of the scatter and rendering the velocity data less reliable. During the quiescence, excess velocities remain low, and the variability is linked to seasonal speed-ups, which are most notable in the main trunk. During the surge, the upper and main trunk, and the

terminus show very similar behaviors, although with a difference in intensity. On the lower end, the driving stress versus excess velocity relationship, shows a monotonic increase until a threshold is reached at which excess velocities continue increasing regardless of driving stress. On the upper end, driving stresses are the highest at relatively low excess velocities and these driving stresses appear to decrease with increasing velocities.



The two types of behavior can be characterized with the generalized slip relationship (Eq. 6) and we pick two sets of
parameters $u_t$, $\sigma_{max}$, $p$ and $q$ (Fig. 8 and Tab. 3) that can encompass the data. Note that these sets of parameters are likely
non-unique and because of the large uncertainties on bed and surface elevation, and the lack of force balance calculations,
we keep this parameter assessment qualitative. Furthermore, Fig. 8 (see also Figs. S8 and S9) highlights that the parameters
of the sliding relationship vary in both space and time, rendering the statistical determination of a single set of parameters
un-physical. In future studies where these quantities are better quantified, however, a robust statistical determination of the
sliding parameters would be extremely valuable.

Interestingly, it appears that a range of parameters is necessary to properly encompass the whole data set (Fig. 8). This range
of value is within the bounds of proposed parameters in the literature (Gagliardini et al., 2007; Iverson, 2010; Zoet and Iverson,
2015, 2020). The range can be explained by different factors. (1) The maximum resistive stress can change locally with bed
properties, for example with maximum bed slope or presence of clasts, in particular of different sizes. We furthermore expect
that a range of obstacle sizes and shapes are present at the glacier bed. (2) Changes in N offset the relationship on the Y-axis
as they control the $\sigma_{max}$. N is expected to change significantly during the surge (Fig. 7b) and accounts for a significant fraction
of the vertical scatter, however, changes in N will only marginally affect the shape of the curve, through the influence of N
on $u_t$. (3) It is realistic to expect that $u_t$ is variable across the glacier as bed characteristics are expected to be anisotropic. (4)
Values of $q > 1$ mean that the slip relationship shows rate weakening behavior after the maximum resistive stress is overcome.
The necessity of the upper bound on the slip relationship to have $q > 1$ means that weakening of the bed resistance takes place
during the surge due, at least partly, to a rate-weakening effect. This can be explained for a rigid bed if sliding is fast enough
to create cavities that engulf maximum bed slopes (Fig. 1); highly likely given the velocities observed. For a deformable bed,
the rotation of clasts and changes in till matrix caused by deformation during the surge, combined with changes in effective
pressure will affect the critical strength of the till (Zoet and Iverson, 2020). However, in the published literature to date, the
proposed sliding relationships for till only consider $q = 1$ (Tsai et al., 2015; Minchew et al., 2016; Joughin et al., 2019; Zoet
and Iverson, 2020).

### 6.5  Implications for glacier sliding

To date, attempts at validating a sliding relationship remain scarce and highlight the caveats of Weertman-type relationships
(e.g. Bindschadler, 1983; Minchew et al., 2016; Vallot et al., 2017; Stearns and Van der Veen, 2018; Joughin et al., 2019).
Minchew et al. (2016) use a combination of remote sensing and numerical modelling to show that sliding at the bed of Hof-
sjökull ice cap, Iceland, follows a plastic behavior. In a study combining numerical modeling and remote sensing to explore the
dynamics of Kronebreen, a fast-flowing tidewater glacier in Svalbard, Vallot et al. (2017) find that the parameters of a sliding
relationship should evolve in space and time to account the behavior they observe. For fast ice flow at Pine Island glacier,
Antarctica, Joughin et al. (2019) show that a sliding relationship involving a Coulomb-type failure is necessary. Similarly,
Stearns and Van der Veen (2018) shows rate-independent shear stresses for several Greenlandic outlet glaciers. These studies
have in common that they assume a deformable bed, and that they are using mostly spatial changes, not timeseries.





Our dataset combined with the generalized sliding relationship allows us to bridge the gap between relatively slow glacier flow, with velocities comparable to Iceland during the quiescence, and fast ice flow, with velocities comparable to that of Greenlandic or Antarctic outlet glaciers. For example Stearns and Van der Veen (2018), looking at 140 outlet glaciers in
Greeland, find no relationship between velocity and basal shear stress, yet they find a correlation between velocities and their proxy for effective pressure. We suggest that these glaciers are undergoing a skin friction regime. The generalized sliding relationship further shows that the maximum bed resistance is mostly controlled by the effective pressure, and the relationship between velocities and basal shear stress in their dataset may exist via the effective pressure, although it is non trivial. Re-analyzing their data in light of the generalized sliding relationship could thus help quantify the free parameters and gain
valuable insight about the bed conditions of these outlet glaciers. In agreement with the findings of Vallot et al. (2017), we find that the parameters of the sliding relationship likely change temporally, thus future quantification should be done while capturing a range of dynamic behaviors for a single glacier.

The results we present suggest that it is possible to test and quantify generalized sliding parameters globally. These efforts are facilitated by the widespread availability of glacier surface velocities (e.g. Gardner et al., 2019). For such quantification
to be efficient though, it is essential to focus on improving glacier bed geometry measurements or estimations. Numerical modeling is also necessary to constrain the stress balance and isolate basal shear stresses (Minchew et al., 2016; Vallot et al., 2017). In addition, the remote sensing method we present is key to identify sub-seasonal surface velocity fluctuations, shedding new light on global glacier dynamics.

### 6.6   Outlook on surge behavior

The generalized sliding relationship highlights the importance of effective pressure in determining both sliding velocity and the maximum bed resistance, regardless of the type of bed. Water pressure fluctuations greater than $10^5$ Pa ($\sim 10$ m of water) are ubiquitous, even for small valley glaciers (e.g. Iken, 1981; Rada and Schoof, 2018), and Kamb et al. (1985) reported fluctuations greater than $10^6$ Pa during the surge of Variegated glacier. On the other hand, till strength for a given effective pressure shows little fluctuations (Iverson, 2010). Over a rigid bed, variations in bed resistance arise from changes in bed geometry and ice
flow-law coefficient (within parameter $A_\mathrm{s}$). Once the transition or skin friction regimes are reached, ice geometry becomes secondary (Fig. 1), and fluctuations in ice flow-law coefficient remain small. The ice at the bed is expected to be temperate, i.e. at the pressure melting point, with little temperature fluctuations, the flow-law coefficient should barely change (Cuffey and Paterson, 2010). Effective pressure thus largely controls $\sigma_\mathrm{max}$ across all bed types.

The effective normal stress, $N$, which depends on the ice thickness and basal fluid pressure, might be seen as a state variable
that controls both the onset of unstable sliding, i.e. the transition and skin friction regimes, but also allows the recovery from fast sliding to occur (Fig. 7b). This means that both the surge initiation and termination can be hydraulically triggered. The idea that enhanced driving stresses lead to initiation or that draining of glacier ice and resulting decrease in driving stresses can cause the surge to terminate would require rates and amplitudes of driving stress changes that are unrealistic. Changes in glacier surface akin to those we observe over two years (2017 and 2019 DEMs) would be necessary to bring the sliding regime





into the transition regime or terminate the skin friction regime, under constant hydraulic conditions. Expecting these changes to occur over the timespan of surge initiation or termination is unrealistic.

The generalized sliding relationship also offers a possible explanation for differences in surge dynamics, for example whether the velocity increase is gradual or sudden. For bed conditions where the threshold velocity is relatively large and $q = 1$, the achievement of skin friction will be gradual and the transition regime will span a large range of sliding velocities. This would

result in a gradual increase in velocities towards the surge apex. Conversely, a relatively low threshold velocity and large $q$, would mean that the transition regime only span a small range of velocities, and the skin friction regime is highly rate-weakening. The surge apex would be reached quickly and surge velocities could remain particularly high until a significant water drainage events occurs.

## 7 Conclusions

We present a generalized sliding relationship for glaciers valid for both rigid and deformable beds, together with a remote sensing workflow that allows us to constrain this relationship from time series of optical images. The new remote sensing workflow allows us to combine surface velocity timeseries calculated from two different satellite and achieve a high spatio-temporal resolution. The combined dataset is then enhanced by post processing steps that significantly improve data quality. We apply this remote sensing workflow to surge-type Shisper glacier in the Pakistani Karakoram and obtain a particularly

detailed picture of the dynamics changes leading up to and during the recent surge. We find an increased amplitude of spring speed-up in the few years leading up to the surge. Then the surge can be decomposed in a slow onset and three main phases. Importantly, these events appear to be linked to the hydraulic activity of the glacier. We thus suggest that the surge is a state of heightened velocity and sensitivity to hydraulic changes.

Our estimation of driving stress and sliding velocity can be encompassed by the generalized sliding law, but a range of

parameters is needed to explain the range of behavior we observe. The most striking outcome is the confirmation of the presence of a transition and skin friction sliding regime during the surge, while the quiescence appear to be characterized mostly by a viscous drag regime. The spring speed-up events likely represent incursions into the transition regime. Our results furthermore suggest that, even for a single glacier, a range of sliding relationship parameters are needed through time and likely for different region of the bed. The present study presents a proof of concept for a method to test glacier sliding relationship

at a global scale, using mostly open-source data and software. We anticipate that reproducing such a study for a glacier with a good bed map and constraint on the ice surface evolution, combined with numerical modeling could yield a quantification of the sliding relationship parameters. Such studies will be essential in better understanding glacier surges and the fast flow of tidewater glaciers in Patagonia, Alaska, Greenland and Antarctica.

*Code and data availability.* The code for image correlation is available here http://www.tectonics.caltech.edu/slip_history/spot_coseis/download_

software.html. The dataset and code used for the analysis and data plotting is available here https://doi.org/10.5281/zenodo.4624397.





## Appendix A: Algorithm

---

**Algorithm 1**    a M t a e te a thm
**Input:** d a ementMa n Ma
**Parameters** , $f, s$
**Output:** d a ementMa

---

**for** *(x,y) in DIM(displacmentMap)* **do**
                                                      `// DIM: image dimension`
    **if** $sn$    $p\,x,y\ \ge T\ n$  **then**
        $sp$    $n$    $p\,x,y$    $N\ N$
        $sp$    $n$    $p\,x,y$    $N\ N$
  $sp$    $n$    $p$    $sp$    $n$    $p \times sn$    $p$
**for** *(x,y) in DIM(displacmentMap)* **do**
  $s$        $0$
  $w$    $sp$    $n$    $p\,x,y,s$
        $n(w)$
    $un$        **CountValid**$(w\ T$     $)$

  **while** $s$        $f$ **do**
    **if**   $un$      $\ge 0.7$ **then**
      $s$        $f + 1$
      $w$    **LocalFilter**$(w\ T$     $)$

    **else**
      $s$        $f$
      $w$    $sp$    $n$    $p\,x,y,s\times$   a e a t
      Med      $n(w)$
        $un$        **CountValid**$(w\ T$     $)$

    $sp$    $n$    $p\,x,y,s$    $w$

**Function** `CountValid`$(w, T$    ,    $)$:
  **for** *(i,j) in DIM(w)* **do**
    **if** $w$ ,    $\pm|T$    $\times$    $|$ **then**
      $un$     $+$    $1$
  $un$        $\frac{un}{(\ )}$
  **return**    nt a d
**Function** `LocalFilter`$(w, T$    ,    $)$:
  **for** *(i,j) in DIM(w)* **do**
    **if** $w$ ,    $\pm|T$    $\times$    $|$ **then**
      $w$ ,      $N\ N$
  **return** $w$

---

**Figure A1.** Algorithm for the post processing of displacement maps.





## Appendix B: Notation

**Table B1.** Summary of symbols

| Symbol | Meaning | Value / range | Unit |
|---|---|---|---|
| $\tau_{\mathrm{b}}$ | Basal shear stress | | Pa |
| $\tau_{\mathrm{d}}$ | Driving stress | | Pa |
| $\sigma_{\max}$ | Maximum resistive stress | | Pa |
| $u_{\mathrm{b}}$ | Sliding velocity | | $\mathrm{m\,s^{-1}}$ |
| $u_{\mathrm{t}}$ | Threshold velocity | | $\mathrm{m\,s^{-1}}$ |
| $u_{\mathrm{ex}}$ | Excess velocity | | $\mathrm{m\,s^{-1}}$ |
| $u_{\mathrm{surf}}$ | Surface velocity | | $\mathrm{m\,s^{-1}}$ |
| $\overline{u}_{\mathrm{quiesc}}$ | Mean quiescence velocity | | $\mathrm{m\,s^{-1}}$ |
| $A_{\mathrm{s}}$ | Sliding parameter without cavity | | $\mathrm{m\,Pa^{-n}\,s^{-1}}$ |
| $p_{\mathrm{ice}}$ | Ice pressure | | Pa |
| $p_{\mathrm{w}}$ | Water pressure | | Pa |
| $p$ | Power law exponent | $2-5$ | - |
| $q$ | Friction law exponent | $\geq 1$ | - |
| $N$ | Effective pressure | | Pa |
| $C$ | Friction law maximum value | $\leq$ maximum adverse bed slope | - |
| $C_{\mathrm{s}}$ | Sliding coefficent | $2.5 \times 10^{-9}$ | $\mathrm{m\,a^{-1}\,Pa^{-p}}$ |
| $\chi$ | Normalized velocity | | |
| $\alpha$ | Coefficient for friction law exponent | | |
| $\phi$ | Friction angle of the till | | |
| $\rho$ | Ice density | | $\mathrm{kg\,m^{-3}}$ |
| $g$ | gravitational acceleration | | $\mathrm{m\,s^{-2}}$ |
| $h_{\mathrm{ice}}$ | Ice thickness | | m |
| $\nabla S_{\mathrm{ice}}(x,y)$ | Ice surface gradient | | |

*Author contributions.* J.-P.A. initiated the project, coordinated the work and contributed to the remote sensing analysis and the interpretation of the results. F.B. identified the study site. All authors framed the research and formulated the goals. S.Aati developed the remote sensing methodology and processed all the data. F.B. helped validate the velocity maps. S.Aati generated the figures presenting the remote sensing analysis and results. F.B. analyzed the remote sensing data and created the result figures. F.B. laid out the generalized sliding relationship, created the conceptual figures and conducted the assessment of the sliding relationship. F.B. led the writing of the manuscript. I.D. assisted in interpreting the processes surrounding the glacier surge and preparing the manuscript. All authors edited the manuscript.






*Competing interests.* The authors declare to competing interests.

*Acknowledgements.* The authors are very grateful for Shan Gremion's help in early stages of the project. F.B. would like to thank Victor C. Tsai and Alex S. Gardner (and many others) for the joint Caltech-JPL group discussion on glacier sliding that greatly helped formulate some of the ideas in this paper. F.B. is grateful for the funding from the Swiss National Science Fund Mobility Postdoc Fellowships (Grants P2SKP2_171755; 2017–2019 and P400P2_191105;2020–2021). We thank the Caltech/JPLPresident's and Director's Fund program for support (Grant 105275-18AW0058;2018).



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
