# Peer review of "Surge dynamics of Shisper Glacier revealed by time series correlation of optical satellite images and their utility to substantiate a generalized sliding law"

_The Cryosphere, 2021_

## Referee Comment (RC2)

**Review of *"Generalized sliding law applied to the surge dynamics of Shisper Glacier and constrained by timeseries correlation of optical satellite images"* by Flavien Beaud et al.**

This study develops an imagery processing pipeline to improve coverage and quality of velocity observations (derived by combining Sentinel-2 and Landsat 8 imagery) of a surging glacier (Shisper Glacier in the Hunza Valley) and uses these detailed observations to 1) characterize the conditioning and trigger of the surge and 2) use the range of velocities observed during the surge cycle to validate a generalized sliding relationship. The velocity processing timeline appears robust, generating realistic velocity fields, and was certainly a large task in and of itself to achieve. The manuscript is well referenced and well-written, and I appreciate the detail taken to explore multiple facets of observed and/or inferred surge characteristics. Beyond the pipeline and interpretation of surge observations, the remainder of the manuscript is somewhat weaker, and could be improved if some of my comments below are addressed or considered. Mainly, discussion surrounding known uncertainties (ice thickness) is lacking, as are quantified evaluations of how well the relationship fits within bounds or as parameters/DEMs vary, which weakens the main argument that the generalized relationship works and is physically sound and useful for understanding surge dynamics. This work is important and can potentially offer the community new tools and insight into glacier sliding, but I think several points of clarification and additional statistics need to be included prior to meeting publication standards.

**Major comments:**

The paper is quite long and does not read as three discreet parts as mentioned in the introduction (although the abstract describes the paper as having two parts). My impression is that much of the orthorectified imagery processing and velocity map development would be better suited for the supplement, with the main manuscript body focused on describing the surge and suspected drivers, and then applying those observations to test the generalized sliding relationship. However, the main tasks of the pipeline could be summarized in the methods with mention of how this pipeline has significantly improved velocity estimates. However, I will leave this up to the discretion of the authors and editor.

My main critique of the paper is that the uncertainties surrounding estimated driving stresses are not thoroughly evaluated (although they are mentioned), making it difficult to assess the accuracy/performance of the generalized sliding relationship. I think more quantitative metrics need to be used to illustrate the sensitivity of results to tuning parameters. Uncertainty in ice thickness stem from both (1) surface elevation uncertainty (DEMs) or (2) uncertainty in the bed topography.
-With regards to (1), excess velocity vs driving stress relationship is said to be mostly insensitive to DEM selected, with results for SRTM and 2019 DEM shown separately in the supplement. However, these comparisons are only qualitatively given. What % of observations fall outside the bounds during these cases compared to the SRTM/2019 DEM mixed assessment? Is a certain

quadrant of the glacier more prone to observations that fall outside of these bounds? Several statistics or quantitative values here would be useful.

-With regards to (2), the text on page 180 reads *"To constrain the bedrock topography of Shisper glacier, we used the three different bed elevation models proposed in Farinotti et al. (2019) and averaged the three results, as suggested by Farinotti et al. (2017)."*

More information here is warranted considering how large an impact the ice thickness imparts on driving stress. What was the range in modeled ice thicknesses compared to mean thickness? Which 3 models were used and why? Were the velocities used for the bed thickness inversion taken from before, after, or during the surge? I do think it is worth calculating changes in effective pressure vs velocity relationship using two end member modeled ice thickness fields in order to show (and quantify) the sensitivity of the relationship to unknown bed topography.

How were the upper and lower values of sigma_max (maximum resistive stress) selected? The upper bound of 500 kPa falls well outside of previously published literature as per Table 3. It seems that this value was selected in order for the upper bound to "cap" the observations and encompass the majority of observed variability. However, how can results showing observaitons generally fall within curated bounds created using parameters tuned to fit the observations a robust way to validate the generalized relationship? Perhaps I am overlooking something, but this seems like circular reasoning to me, and some further clarification would also aid readers.

**Minor comments**

Section 5.4 – mass balance of Mochowar glacier
The inclusion of this section is confusing without a companion section for Shisper glacier. The most important elements of this section can probably be integrated more seamlessly into sections 5.1 and 5.3.

I am interested in the lack of evident fall speed up in 2017 (based on Figure 5), preceding the onset of the surge in winter 2017. Can you comment on how its absence relates to the broader picture of hydrology-driven surging?

Table S1 – it seems that in 2016 and 2017, the time separation between pairs can be quite large, reducing the temporal resolution of summer to autumnal velocity measurements. How might this impact the ability to constrain the timing and magnitude of a "fall speed up"?

Section 6.2 Surge Trigger
I agree with the authors that there is sufficient evidence that surging at Shisper glacier is hydrology, rather than thermally, driven. However, I would like to see the manuscript comment on some reasons why the surge initiated *when* it did (as in, why specifically in late 2017, early 2018 rather than an earlier year). Is this simply due to increasing magnitudes of spring speed up in 2016 and 2017?

Line 240, regarding the data cube, D. I am having trouble following how images are grouped according to an image target date. Are all images that fall within a specific temporal window of the target date included in the stack?

Line 490 – correct spelling for "Greeland"

There are many incomplete/missing sections in the supplement. These will be filled in the next iteration of the manuscript?

---

## Author Comment (AC1)

Answer to reviewers "Generalized sliding law applied to the surge dynamics of Shisper Glacier and constrained by timeseries correlation of optical satellite images" by Flavien Beaud et al."

We would like to thank the three reviewers for their time and constructive comments. Before addressing each reviewer's comments independently, we summarize our understanding of the main criticisms and how we plan to address them in the revisions.

We are glad to see that the reviewers made an overall positive assessment of the manuscript. The section that draws substantial criticism is one sub-section of the discussion, section 6.4. In this section we assess whether the proposed generalized sliding relationship can encompass the data presented. Reviewer 1 and 2 both express concern that the uncertainties related to the bedrock topography, ice surface elevation and basal stress estimations are too large to trust the assessment of the generalized sliding relationship. Our goal in that section is to present a first order assessment that the proposed generalized sliding law could explain the observed data, and that it is possible to constrain sliding-law parameters in cases where estimates of surface and bedrock topography are available. We note that the the concerns raised by the reviewers are the very reason why we avoid quantifying the sliding-law parameters. We also agree that the data scatter originates from both heterogeneities in bed properties and measurement uncertainties, but the signal is large enough so that our qualitative assessment holds.

Reviewer 3 (D. Benn) suggests some re-framing of the sliding law that would make the role of effective pressure (N) explicit. We explain below that we didn't follow this advice because it would then require choosing between a soft or a rigid bed from the onset and the formulation would then loose generality. We, however, propose changes to the text that better describe the importance of effective pressure in the generalized law and should better substantiate our interpretation of the surge data.

We also futher explain how the bed data estimates from Farinotti et al. (2019) has little effect on our main conclusion. We propose to add figures to the supplementary material to show how using different surface DEMs or bed elevation estimates affect the sliding-law relationship. These figures (see new figures) show that while the set of parameters that best fits the dataset might change, the proposed sliding law could still explain the observations. Given the difficulty to assign realistic uncertainties to the bedrock elevation, we cannot perform a quantitative analysis of the uncertainties on the sliding-law parameters.

Below are the answers to the specific comments for each reviewer and afterwards, the proposed changes to the text. Reviewers' comments are in black, our answers in blue, and proposed changes in green.

**Answer to Reviewer 1**

Review for:
"Generalized sliding law applied to the surge dynamics of Shisper Glacier and constrained by timeseries correlation of optical satellite images" Beaud et al.

Summary:
The authors present a study with the overall goal to use satellite observations of a surging glacier to constrain the form of a generalized sliding law. The manuscript is overall very well written, especially the first half, and the figures are informative and of high quality. The manuscript has three main parts: First, the authors derive a new generalized form the traction relationship which combines rate-strengthening with weakening applicable to both hard and deformable beds. The second part presents satellite velocity and elevation observations of a glacier surge captured on Shisper Glacier. Finally, the last part uses these observations to constrain parameters within their generalized sliding law presented in the first section.

The goal of the study is very worthy. We have much work to do to better understand basal physics and I am generally enthusiastic about the first two sections. I quite like the generalized relationship derived in the manuscript, especially when compared to less physically based generalized sliding laws. However, I still find little advantage for transient simulations where defining $u_t$ still requires aprior knowledge of the bed conditions. For these sections, I include some suggestions to improve these sections detailed in the lineby-line.

Thank you. It is unclear what the reviewer refers to as simulations. We make several approximations (discussed late by the reviewer) but there is no numerical modeling involved in the current study.

However, I have fundamental concerns about the methods/data used to constrain the traction relationships presented in the last section.

Thank you for your overall positive comments on the manuscript.

- Investigating traction relationships require the traction and velocity to be very well constrained. In this regard, the basal shear stress is the biggest concern. The methods to invert for the topography presented by Farrinetti 2019 use simplified physics that assume all motion is derived from ice deformation. Given the observations presented in the paper and the hypothesis that the glacier responds to melt suggests that this is not the case. Thus, while the variations in topography derived from this method

likely reflect variations in the bed topography, the actual thickness of the glacier in the paper is largely uncertain. On top of this is the fact that the glacier is surge type, so inverting for the bed topography will also have a large dependency on what velocity field or surface elevation data set is used for the topography inversion. The authors do not note what data they use for the inversion, although this is not the main issue here.

We did not determine the bed topography ourselves, we used the product of Farinotti et al., 2019 and all the details about the model and uncertainties can be found therein. We will make this point clearer in the revision. The main product is one estimation of bed elevation based on the composite of 3 different models for Shisper glacier.

We agree that the uncertainties are large and difficult to estimate, hence our choice of a qualitative interpretation. We tested different bed and surface topography models, the results show that the signal is large enough that the main conclusions hold regardless of the bed elevation model or surface elevation model used. The proposed changes should make this point clearer in the manuscript.

The reviewer's statement about the methods presented in Farinotti et al. (2019) is not exactly correct. Two out of the three models (models 1-3 in Farinotti & 2019) include a parametrization of basal sliding while assuming simplified physics and the 3rd is mostly empirical. These models have been applied to marine terminating glaciers and their validity is thus not limited to a deformation-dominated flow scenario.

- Estimating the basal shear stress using the driving stress for a glacier undergoing large transient forcing is insufficient. In the case of a surging glacier, sharp and variable gradients observed in the velocity field indicate higher order stresses will undoubtedly play a role in the basal traction field. This requires more sophisticated inversion methods such as the SSA or Full Stokes which take into account higher-order stresses by using a more complete formulation of the momentum balance. However, these methods are only as good as the data that constrain the inversion, in which case there are still problems, one of which I have outlined above, another is knowing the ice rheology, and the last and probably the most important I will outline below.

Using the driving stress to estimate the basal shear stress will lead to an overestimation of basal shear stress (see Minchew 2016, Thogersen 2019, etc.). That means that if we can show rate independent or rate-weakening behavior, that would only be further confirmed by a better quantification of basal shear stress.

Proposed changes L434 to 435 from:

We then plot an estimation of the relationship tau_b versus u_ex for our dataset (Fig. 8).

To

We choose to use the driving stress (tau_d) as an approximation for the basal stress (tau_b). While this assumption is coarse, it ensures that we overestimate the basal stress at large sliding velocities (see Minchew 2016, Thogersen 2019), making it more difficult to observe rate-independent or rate-weakening behavior. We then plot the relationship tau_b versus u_ex for our dataset (Fig. 8).

- A basic feature of a surge is large is transient surface geometry changes (i.e. Kamb et al. 1985) which will have a significant impact on the stress field. Evidence of large surface geometry changes is presented in the paper, where 100s m ice thickness changes can be observed through the three elevation data sets presented. However, comparing the velocity field which represents a snapshot for a specific time period during a surge to an elevation dataset averaged overtime, we have no idea and really no way of determining whether the stress field is consistent with the velocity field at that point in time.

That is correct. There are figures, including new ones, in the supplement showing the different results with different surface elevations. We have added the figures showing the tau_d vs u_b relationship with the different bed elevations estimated by Farinotti et al, 2019.

The plots for all the bed and surface elevation models available show similar patterns and, importantly the rate-independent or rate-weakening behavior. We thus argue that our quantitative conclusions are robust

We have also added some discussion about the relative effect of different variables on sliding in the newly rewritten discussion subsection (section 6.5 shown at end of response). In summary, changes in effective pressure are expected to be at least as large and up to an order of magnitude larger than changes in driving stress, but can occur over much shorter timescales.

With the combined effect of these three sources of uncertainty for the traction field, all of which as presented are nearly impossible to assess, I do not see how conclusions regarding traction relationships can be made unless other high resolution elevation data sets could be found that match the time period of the observed velocity data and better inversion methods are used.

We made it clearer in the revision that we did not attempt a formal and robust inversion for the sliding-law parameters due to the uncertainties that the reviewer idenfified. We are proposing a framework that could allow quantification, and point to what future study should do to take this to the next level.

A revised manuscript would need to address this prior to be considered for publication. The challenges for the last section are considerable. However, the data and interpretation are interesting and in theory I do like the idea of trying to use surge behavior (with better data) to populate a traction curve (although there are a lot of things about surges (i.e. heavy crevassing) that make inverting for the traction field difficult.) A revised manuscript might want to focus on the surge behavior where a discussion on the potential for constraining traction relationships using surge glaciers would surely be interesting.

Thank you for your enthusiasm about the general goal of the study. The revised manuscript will make it clearer that the section of concern is a first order assessment and by no means an attempt at a robust quantification, and that it's use is promising where better data is available.

Specific Comments

Line 20:21: This generalization is not necessarily true. There are direct observations at several locations in Greenland rebuke this paradigm (i.e. Ryser 2014, Maier 2019 – measurements slow flow, Lüthi 2001, Doyle 2018 – measurements in fast flow).

Thank you. We will change the wording from "fast" ice flow to sliding-dominated ice flow throughout the manuscript and regarding these lines we will make the following changes at L. 21-23 from:
Ice flow velocities exceeding one meter per day are considered 'fast', and correspond to an ice flow regime that is dominated by basal sliding rather than ice deformation. Fast ice flow at tidewater glaciers is responsible for the majority of mass loss from the Greenland Ice Sheet […]
To:
Sliding-dominated ice flow at tidewater glaciers is responsible for the majority of mass loss from the Greenland Ice Sheet […]

Line 28-30: This is not likely generalizable everywhere. See (Maier 2021) for traction analysis that suggests some fast flowing regions generally obey rate-strengthening.

The focus of this study is on the sliding-dominated ice flow, not just 'fast' ice flow. Thanks to the reviewer comment, we now corrected the text to reflect that.

Proposed changes L. 27-28, from:
Fast ice flow velocities cannot be adequately explained by traditional Weertman-type sliding relationships […]
To:
Large sliding-dominated ice flow velocities cannot generally be explained by traditional Weertman-type sliding relationships […]

Line 31: Strange phrasing here.

L. 31 reads: "The idea that bed shear stress is bounded for large sliding velocities was proposed in early physical glaciology work […]"
It is unclear to us what is strange in this wording.

Line 111-114: This description is a bit confusing. I am pretty sure skin friction is the same as viscous drag for a non-turbulent boundary layer. This needs to be much better defined.Right now I think you are actually referring to solid friction (i.e. generated from debris-bed interactions)? You need make it clear that this is what you are referring to, as this is the only thing that makes sense for the bottom panel of the hard bed figures in Figure 1.

Skin friction can be viscous drag at the boundary layer scale, but it is form drag at the macroscale. We will replace 'viscous drag' here by 'form drag' as it is more accurate for our current description.

Also while the could the case is made that there is a physical transition regime for till beds, can the same be made for hard beds? Based on your plots on Figure 1 the transition regime seems arbitrarily drawn after rate-weakening begins.

The presence of that transition is what was shown by Iken early on and validated by Schoof (2005) and Gagliardini et al. (2007). The beginning of the transition zone is defined by $u_t$ in both rigid and deformable beds, while the end is indeed arbitrarily drawn as a concept. We will make that clearer in the figure caption.

I would think the transition regime occurs between the start of cavitation and Iken's bound.

That is true. We modified figure 1 and its caption to show that the transition regime will be different for different values of q. [MODIFY FIGURE]

Line 100: There is probably a more direct way to say this. Also see Helanow et al 2021 for the latest on numerically derived sliding laws over realistic beds. This might help your case that there is no need to explicitly model rate-weakening.

L100-103 read: "The bed of a glacier can only produce a finite amount of resistance to flow that is determined by its properties (CITATIONS) challenging the validity of Weertman-type relationships which imply an unbounded resistance to flow."

This sentence is meant to be general across all bed types, and we purposefully consider the possibility of rate-weakening behavior, as we find our results to be supportive of that behavior. The paper mentioned is focused on hard beds.

Figure 1: Ok the skin friction here is much better defined in a conceptual sense, but you are talking about solid friction, and this is not included in the sliding law of Gagliardini 2007. Iverson 2003 provides some of the only direct measurements of solid friction, however, it has not been incorporated nicely into sliding laws nicely yet in a theoretical sense. Some nice experimental work on solid friction was also done recently by Thompson et al. 2020. These papers might help better formulate your "hard bed skin friction" regime.

We are not talking about solid friction, we are talking about skin friction as ice is treated as a fluid and deforming till can be considered as such as well. Skin friction is the viscous drag at the scale of boundary layer, between rock and ice. We will clarify that in the legend of figure 1 and will further replace viscous drag in the text by form drag, which here is the macroscopic viscous drag due to bed obstacles. The terminology is the same as in Minchew and Joughin (2020).

135: I like the concept of the generalized law you propose Eq 6., but the utility for me is somewhat lost. To model transients you still need to know the effective pressure and thus all the bed specific parameters in Eq 5 right? I think explaining the benefit of your generalization would be a nice thing to add here.

The interest of having a generalized sliding relationship of all glaciers is that, while analyzing observations of surface velocities, one can use one single equation without having to make an educated guess as to what the bed type might be. We don't propose a new mechanism, merely a unification of the existing ones. The possibility of single equation is the title of the reviewer highlight of Zoet and Iverson (2020) by Minchew and Joughin (2020). There is no modeling done in the current study, but we lay a road map for future modeling studies to improve and constrain the parameters.

177-181: What is the uncertainty on the bed and thickness? This is fundamentally important to estimating the basal shear stress.

We refer to Farinotti et al. (2019) for the specifics regarding the methods. The final product is a composite fo the different models used that yield a solution. Creating that composite is yield the best approximation. They didn't provide uncertainties but they provided solutions obtained with the 3 different models. We tested these different solution and found that in all cases the data could be reconciled with the proposed friction law.

236-240: Using a PCA to reconstruct a timeseries needs much more explanation. Why do you do need to do this? What is the advantage? Is the data really noisy?

We will clarify the text in that paragraph as it appears that the text is misleading. The PCA is used to construct the spatio-temporal pattern of surface velocity most consistent with the entire dataset. The advantage is that it reduces the noise while preserving the spatial resolution (contrary to a convolutional filter), and produces more consistent velocity maps. For completeness the timeseries of velocity before the reconstruction with PCA have been shown in the supplement. Sources of noise and the effect on data is discussed in the results section.

Changes 238-239 from:
To identify these correlations, we use a PCA and reconstruct the time series with the limited number of components needed to recover the data variance.

To:

To identify these correlations, we use a PCA and reconstruct displacement maps with the limited number of components needed to recover the data variance. This process reduces the noise level while preserving the inherent spatial resolution of the data.

Figure 4: Interesting figure. I would possibly try to regroup this to show surge and nonsurge behavior to make it to really emphasize the difference between the two states. If there is any way to consolidate into less panels this might help make things clearer. Maybe bold font for the inset text boxes? Really hard to read.

The point of this figure was to show different velocity patterns in chronological order, and to show all the important patterns necessary to understand the dynamic phases of glacier surging. The text inside the figure is already in bold or medium font to ensure better viewing

270-275: Fall speed up is quite interesting. Any idea why? Stress increase from snowfall?

It is likely a Fall-speed up as observed in other glaciers linked to changes in subglacial hydraulogy changes. We describe this in the discussion (last paragraph of 6.2 Surge trigger in original MS). Also, because of lack of data, we cannot go too deep into the interpretation.

283: You can also define based on propagating surge bulge seen in the strain field? To me the surge is hard to identify as a propagating velocity wave in Fig. 4 or 5.

It is, in theory, possible, but would still rely on an arbitrary definition. We tried to qualify a surge bulge evolution and surge propagation velocity, but the data is still not at a high enough spatio-temporal resolution to do so.

Fig 5: Also an interesting figure. What is your reference for along flow distance (i.e. what side is the terminus?)? It would also be cool if you put your interpretation of the 'surge front' at different locations here (maybe with some accompanied interpretive text). It isn't super obvious where this is from your velocity maps, and also might make for interesting discussion (maybe near lines 297-300).

We will add kilometer markers on Figure 2 to show the locations We will clarify that the left side of the plot is the top of the glacier.]

We don't think that a detailed mapping of the terminus is necessarily relevant for the current study, as it wouldn't change the interpretation. The advance of the terminus is plotted in the third panel of Figure 5. This mapping has also been done in different manners in a few studies already (See other papers on Shisper).

311: Did you visually check for artifacts?

Not systematically. The processing is done automatically by the algorithm, that is one of the purposes of its design.

315: Can you indicate the location of these regions on Fig. 5?

Yes, the figure will be updated to show the different zones.

345: Can the fact that these glaciers were recently connected explain some of the changes in geometry?

This is a good question. The rapid demise of the Mochowar is clearly due to inherited ice geometry. The expected changes would be limited to the dead ice zone of Shisper left after

the surge. We expect that effect to be very local, having minimal impact on the Shisper's geometric evolution.

358-360: Sentence unclear.

We propose to change the text from (L. 357-360):

We do not have enough information to discuss the differences between the two surges, but it suggest that the fast flow part of a surge cycle can vary over time. The possible difference between surges of a single glacier could challenge the idea that glaciers in different climatic regions have different surging mechanisms (e.g. Murray et al., 2003; Quincey et al., 2015), however suggest that a single mechanism has different manifestations.

To

We do not have enough information to discuss the differences between the two surges, but we suggest that the fast flow part of a surge cycle can vary over time. The possible differences between surge dynamics at a single glacier could suggest that surges follow a single mechanism with different manifestations, rather than different surging mechanisms linked to climatic regions (e.g. Murray et al., 2003; Quincey et al., 2015).

367: So surge here is inferred to only happen in conjunction with melt forcing?

Generally, yes. Although, increase melt is not the only factor, the Fall speed-ups are associated with decreasing surface melt.

386-387: As stated without additional evidence this is highly speculative.

These lines refer to the statement that our data is limited in temporal resolution and that we do not capture hourly or daily velocity changes with the Kamb (1985) citation. This citation is meant to imply that hourly and daily velocity changes can happen during a surge.

412-419: It would be interesting to know when the lake reaches flotation, as this could also induce unstable behavior similar to a surge.

Yes, it would indeed. However, the data quality does not allow us to obtain that information. We also explain the possible influence of the lake formation on the glacier dynamics.

Remainder of manuscript: I have commented enough on the remainder of the manuscript at the beginning of the review. In short, it is hard to do this analysis if the traction field is not

confidently known. While I suspect that some of your conclusions, i.e. there could be a wide range of traction parameters in a small spatial domain, could be true, but its hard to see how you could differentiate a wide parameter space from errors. A few remaining comments:

Table 3: Gillet-Chaulet 2016, De Rydt 2016, and Maier 2021 all have parameters that you can add to this list.

Thank you for these suggestions. While these papers hold interesting information, none provide estimations for the parameters listed in the table with a similar context. The parameters presented in these papers are thus not directly relevant for this table.

Figure 8: Excess velocity is difficult to understand and conceptualize how it relates to a traction relationship. Can you populate a traction curve by just looking at changes in velocity with out regards to the original velocity? I think this would need an illustration of some sort.I

The excess velocity is a simple way to ensure that most the signal used is due to sliding, and not ice deformation. While it does not provide an accurate quantification of sliding velocity, removing the "background" velocity ensure that most the velocity changes observed are attributable to sliding. Thus, that the shape of the relationship will be adequate, although it could be offset along the velocity axis.

The reviewer proposes to look at changes in velocity. That change must be calculated with respect to a reference, which, by definition, will also be arbitrary. Here the reference is the background velocity. The equation showing how the excess velocity is calculated should provide a clear definition as to what it represents.

What sort of area does each dot represent? This is important when determining how independent each grid cell is.

The bin plot bins the data by range of stress / velocity combination. As described in the caption, each axis is divided into 40 bins of stress and velocity and the color represents the numbers of data points that falls in each bin. Each data point is a pixel on a velocity map.

485: The plots presented are also mostly examining spatial variations.

We do not entirely agree with this assessment. Although our figures show temporal changes, the studies mentioned at that line are indeed mostly spatial, which we say in the next line.

Appendix: This is really difficult to understand.

We will add a more detailed description of the content.

**Answer to reviewer 2**

This study develops an imagery processing pipeline to improve coverage and quality of velocity observations (derived by combining Sentinel-2 and Landsat 8 imagery) of a surging glacier (Shisper Glacier in the Hunza Valley) and uses these detailed observations to 1) characterize the conditioning and trigger of the surge and 2) use the range of velocities observed during the surge cycle to validate a generalized sliding relationship. The velocity processing timeline appears robust, generating realistic velocity fields, and was certainly a large task in and of itself to achieve. The manuscript is well referenced and well-written, and I appreciate the detail taken to explore multiple facets of observed and/or inferred surge characteristics. Beyond the pipeline and interpretation of surge observations, the remainder of the manuscript is somewhat weaker, and could be improved if some of my comments below are addressed or considered. Mainly, discussion surrounding known uncertainties (ice thickness) is lacking, as are quantified evaluations of how well the relationship fits within bounds or as parameters/DEMs vary, which weakens the main argument that the generalized relationship works and is physically sound and useful for understanding surge dynamics. This work is important and can potentially offer the community new tools and insight into glacier sliding, but I think several points of clarification and additional statistics need to be included prior to meeting publication standards.

**Major comments:**

The paper is quite long and does not read as three discreet parts as mentioned in the introduction (although the abstract describes the paper as having two parts). My impression is that much of the orthorectified imagery processing and velocity map development would be better suited for the supplement, with the main manuscript body focused on describing the surge and suspected drivers, and then applying those observations to test the generalized sliding relationship. However, the main tasks of the pipeline could be summarized in the methods with mention of how this pipeline has significantly improved velocity estimates. However, I will leave this up to the discretion of the authors and editor.

We reframe the abstract and introduction wording to show that the study has two main goals (highlighted in abstract) but saying there are 4 parts might be more accurate, (1) sliding theory, (2) remote sensing methods, (3) remotes sensing results, (4) discussion of results in light of sliding theory.

Change L. 6-7 in the abstract from
The present study has two parts:
to
The present study has two main goals:

Changes to the introduction that reflect better paper structure and more careful workding about the assessment of sliding relationship: L. 89 -94 from

This paper is composed of three parts. First, we show that rigid and deformable bed theories can be combined into a generalized sliding relationship applicable for any glacier environment and dynamic evolution. Then, we present the remote sensing method, show the results for Shisper and Mochowar glaciers, and use the measurements to characterize the surge of Shisper glacier. Finally, we use the data set collected for Shisper glacier to validate and constrain the generalized sliding relationship and contextualize how the generalized sliding relationship can improve our understanding of surge dynamics.

To:

This paper is composed of four parts: (1) we show that rigid and deformable bed theories can be combined into a generalized sliding relationship applicable for any glacier environment and dynamic evolution. (2) We present the remote sensing method and workflow. (3) we apply this workflow to Shisper and Mochowar glaciers and characterize the surge of Shisper glacier. (4) We use the data set for Shisper glacier to perform a first-order assessment of the generalized sliding relationship and contextualize how this theory can improve our understanding of surge dynamics, and more generally, basal sliding.

NOTE: "validate and constrained" changed to "perform first-order assessment".

We originally thought about splitting this study into two separate papers, one with the remote sensing methods and results, the other with the generalized sliding law and its assessment. However, because of the uncertainty in the assessment, we were concerned it would be too weak for a paper of its own. We believe the image processing is novel and deserves its own part in the main text. We would thus prefer to keep the structure as is.

My main critique of the paper is that the uncertainties surrounding estimated driving stresses are not thoroughly evaluated (although they are mentioned), making it difficult to assess the accuracy/performance of the generalized sliding relationship. I think more quantitative metrics need to be used to illustrate the sensitivity of results to tuning parameters. Uncertainty in ice thickness stem from both (1) surface elevation uncertainty (DEMs) or (2) uncertainty in the bed topography.

We have made clearer in the revision that we are not trying to assess the accuracy of the sliding relationship. Our goal here is merely to show widespread rate-independent or weakening behavior and that sliding parameters are spatially variable. Then we suggest that this method applied to a site with better data can lead to an actual evaluation of the accuracy and performance of this relationship.

Changes made to introduction reflect that. We believe that consistently referring to our work a "first-order assessment" characterizes the qualitative and coarse nature of what we can achieve with the current dataset.

-With regards to (1), excess velocity vs driving stress relationship is said to be mostly insensitive to DEM selected, with results for SRTM and 2019 DEM shown separately in the supplement. However, these comparisons are only qualitatively given. What % of observations fall outside the bounds during these cases compared to the SRTM/2019 DEM mixed assessment? Is a certain quadrant of the glacier more prone to observations that fall outside of these bounds? Several statistics or quantitative values here would be useful.

What is said in the conclusions is that the shape of the relationship remains similar (i.e. rate-independent and rate-weakening behaviors are present) and that a range of parameters is needed to encompass the data. We believe these conclusions can be made qualitatively and are insensitive to the aforementioned uncertainties.
We do not claim the relationship remains the same, and the figures in the supplement show that the best parameter set should be different. As mentioned earlier, we are keeping the analysis qualitative on purpose because the data is not good enough to produce a quantitative validation. We further believe that a more thorough analysis with the current dataset could lead to misleading interpretations, and we prefer to suggest that that can be done in further studies with improved dataset.

-With regards to (2), the text on page 180 reads "*To constrain the bedrock topography of Shisper glacier, we used the three different bed elevation models proposed in Farinotti et al. (2019) and averaged the three results, as suggested by Farinotti et al. (2017).*" More information here is warranted considering how large an impact the ice thickness imparts on driving stress. What was the range in modeled ice thicknesses compared to mean thickness? Which 3 models were used and why? Were the velocities used for the bed thickness inversion taken from before, after, or during the surge? I do think it is worth calculating changes in effective pressure vs velocity relationship using two end member modeled ice thickness fields in order to show (and quantify) the sensitivity of the relationship to unknown bed topography.

We acknowledge that we didn't explain clearly what bed topography was used. The main product from Farinotti 2019 is a composite ice thickness estimation based on 3 different models. The models and their differences are described in details in Farinotti 2017, and the final data product and its creation described in Farinotti 2019. We have tested all the models proposed in that study, but focus on their main product which is the composite model.

Proposed text change at L.181-182 to replace:

To constrain the bedrock topography of Shisper glacier, we used the three different bed elevation models proposed in Farinotti et al. (2019) and averaged the three results, as suggested by Farinotti et al. (2017).

with:

To constrain the bed topography of Shisper glacier, we use the data products from Farinotti et al. (2019). For Shisper glacier, the ice thickness was estimated as the composite of three different models, as detailed in Farinotti et al. (2017). The specific DEM used is a combination of ASTER and SRTM DEMs for which the timing of acquisition is not specifically known. It is likely in the mid-2000s, after the last surge of Shisper glacier.

To show the effect of using different bed topographies, we have added figures to the supplementary material showing the velocity vs driving stress relationship for all three beds available. We have also added a figure in the sup mat showing the different bed elevations resulting from the 3 models and the composite bed elevation.

However, we don't believe we are able to calculate changes in effective pressure, we can merely qualify them.

How were the upper and lower values of sigma_max (maximum resistive stress) selected? The upper bound of 500 kPa falls well outside of previously published literature as per Table 3. It seems that this value was selected in order for the upper bound to "cap" the observations and encompass the majority of observed variability. However, how can results showing observaitons generally fall within curated bounds created using parameters tuned to fit the observations a robust way to validate the generalized relationship? Perhaps I am overlooking something, but this seems like circular reasoning to me, and some further clarification would also aid readers.

This is correct. The parameters were indeed chosen to encompass the data shown, and the data fits within these bounds. This would not be the case if we used a Weertman-type sliding relationship. At lines L 453 to 457 we describe the choice of parameters as likely non-unique and our selection qualitative. Again, we are not looking for a robust validation, we are trying to show a qualitative fit, which can be done visually.

Perhaps some of the text is misleading on L. 461-462 as this was meant for p and q only. We propose to change

This range of value is within the bounds of proposed parameters in the literature (Gagliardini et al., 2007; Iverson, 2010; Zoet and Iverson, 2015, 2020).

To

This range of values for p and q is within the respective bounds of proposed parameters in the literature, and sigma_max and u_t appear realistic (Gagliardini et al., 2007; Iverson, 2010; Zoet and Iverson, 2015, 2020). Only laboratory or numerical experiments are available to compare the latter two parameters hindering a direct comparison.

The reason for the large driving stresses is explained L 443-444:
"The ice fall and the front of the terminus show particularly high driving stresses because of surface slopes greater than 30∘ (Fig. 9). "

**Minor comments**

Section 5.4 – mass balance of Mochowar glacier
The inclusion of this section is confusing without a companion section for Shisper glacier. The most important elements of this section can probably be integrated more seamlessly into sections 5.1 and 5.3.

Section 5.3 describes the mass balance change at Shisper, we believe this is the companion section.

I am interested in the lack of evident fall speed up in 2017 (based on Figure 5), preceding the onset of the surge in winter 2017. Can you comment on how its absence relates to the broader picture of hydrology-driven surging?

This is indeed intriguing. It is important to clarify that the lack of evidence in the current data set for a fall speed up in 2017, doesn't necessarily mean it didn't occur. We already discuss the fact that the nature of the current dataset means that we will miss some aspects of ice flow that occur at shorter timescales than time difference between the pictures (see L70 71: "*Glacier velocities derived from remote sensing will therefore consistently miss sub-monthly fluctuations, in particular those at the daily or sub-daily timescales which can be significant*"). Also see lines 380-389 for further discussion.

Table S1 – it seems that in 2016 and 2017, the time separation between pairs can be quite large, reducing the temporal resolution of summer to autumnal velocity measurements. How might this impact the ability to constrain the timing and magnitude of a "fall speed up"?

That is correct. It can prevent us from seeing velocity speed-ups.

Section 6.2 Surge Trigger
I agree with the authors that there is sufficient evidence that surging at Shisper glacier is hydrology, rather than thermally, driven. However, I would like to see the manuscript comment on some reasons why the surge initiated when it did (as in, why specifically in late 2017, early 2018 rather than an earlier year). Is this simply due to increasing magnitudes of spring speed up in 2016 and 2017?

This is a good question. We have combined two sections of the discussion and added a discussion of the enthalpy / mass budget model as also recommended by reviewer 3.

Line 240, regarding the data cube, D. I am having trouble following how images are grouped according to an image target date. Are all images that fall within a specific temporal window of the target date included in the stack?

Sentence L 241-242 changed from:
Velocity maps are then stacked on the same data cube D according to the target image date, data cube that is decomposed in principal components.

To

Velocity maps are then stacked on the same data cube D according to the older image date. When a velocity map overlaps with the preceding one, we only keep the older velocity map for the time span between the two maps older images. The data cube that is decomposed in principal components.

Line 490 – correct spelling for "Greeland"

Thanks, corrected.

There are many incomplete/missing sections in the supplement. These will be filled in the next iteration of the manuscript?

Yes, we will clarify the supplement.

**Answer to reviewer 3 (Douglas Benn)**

This is an intriguing paper that is rich in both data and ideas. Among other things, it represents a bold and innovative attempt to use remote sensing observations to calibrate a proposed 'generalized sliding law'. There are some shortcomings in the implementation of this idea and the discussion of the results, but with some additional work the paper should become a very original, worthwhile and thought-provoking addition to the literature.

Thank you.

Following a well-written introduction, the paper provides an admirably clear exposition of recent sliding law literature, which culminates in the proposal of a 'generalized sliding law'. The proposed solution is elegant, and neatly highlights the fundamental similarity of existing hard-bed and soft-bed sliding laws. However, the proposed law subsumes effective pressure N into both the threshold sliding velocity Ut and maximum resistive stress σmax, meaning that these two main components of the Ub - Tb relationship are inherently subject to large variations, particularly on glaciers that exhibit large and rapid velocity variations (e.g. surging glaciers). This is reflected in the very large spread of values in Figure 8, which the authors interpret more or less entirely as the result of variations in N. Therefore, I feel that in its present form, the generalized law is not particularly useful.

Thank you again for these positive comments. Regarding the "usefulness" of the current relationship, we believe it is very useful to have one general relationship that can be used in any case rather than having to choose one based on assumptions on bed properties. It is indeed unfortunate that N does stand out more clearly, however, the complexity of the model at play is not a choice and retains that of previous studies.

It would be better to retain N as an explicit variable in the sliding law by replacing Ut with something like a.N and σmax with b.N, where a and b are 'constants' largely determined by bed properties (C, As, tanφ etc.). Although a and b will be spatially variable, this approach might allow temporal variations in N to be approximated if τb and Ub are known, potentially extracting considerably more structure from the data gathered by the authors.

Being able to do so would be ideal and would indeed clarify the interpretation. While it is possible to do so for $\sigma_{\max} = N \tan \phi \; or \; NC$, it is not possible for $u_{\mathrm{t}}$. While $u_{\mathrm{t}}$ is linearly dependent on $N$ for a deformable bed and can be expressed as $aN$, where $a$ is a spatially variable parameter accounting for till properties, $N$ is raised at the power $p$ (Eq. 5) for a rigid bed, and $p$ is comprised between 3 and 5 in the current study. It is thus not possible to write the current generalized sliding relationship while isolating $N$. In the discussion we go into

details as to what could cause changes in either $\sigma_{\max}$ and $u_t$ and identify them as being most likely related to $N$.
What propose to clarify in section 2 that the parameters (except N) used to calculate $\sigma_{\max}$ and $u_t$ likely change spatially, but they are less likely to change in time. Thus, most temporal changes in $\sigma_{\max}$ and $u_t$ are likely the results of changes in N. We also clarify that $\sigma_{\max}$ and $u_t$have very different effect on general shape of relationship.

L. 134 Changes to equation 5 to better reflect the role of N in determining threshold velocity over a defomable bed.

$$u_t = \begin{cases} C^p N^p A_s \\ f(N, bed\ properties) \end{cases}$$

Changed to

$u_t = \begin{cases} C^p N^p A_s \\ C_d N \end{cases}$, where $C_d$ is a spatially variable parameter accounting for bed properties (see Eq. 2 in Zoet and Iverson, 2020),

That way N is expressed explicitly in $\sigma_{\max}$ and $u_t$.

To add on line 136 after Eq.6 (generalized sliding relationship):

From Eqs. 4–6 N stands out as the variable responsible for temporal changes in $\sigma_{\max}$ and $u_t$. The other parameters (C, As, Cd and φ) have been shown to change spatially (Gagliardini et al., 2007; Zoet and Iverson, 2020) but are not expected to vary significantly over time spans relevant for a glacier surge, i.e. a decade. The fact that N is raised to the power p > 1 in Eq. 5, however, precludes from writing Eq. 6 for the general case explicitly as a function of N.

The data on the surge of Shisper glacier are excellent and very interesting. The combination of Sentinel and Landsat data provides a dense and high-quality velocity series, allowing the evolution of the surge to be interrogated in detail. The description of the processing work flow should prove useful to other workers in the field. Given the quality of the data, however, the discussion of the underlying processes is disappointing. The discussion of surge mechanisms presents an out-dated binary choice between thermal and hydraulic switch mechanisms, and by dismissing the thermal mechanism, the authors conclude that the hydraulic mechanism as the only other option. In its widely accepted form, however, the hydraulic switch idea is problematical. First, no convincing mechanism has ever been proposed for why drainage systems should spontaneously switch from channelised to distributed forms (the detailed analysis of Kamb 1987 focuses on the opposite switch, which satisfactorily explains surge termination but not onset; and Fowler's model of the hydraulic

switch simply incorporates an heuristic function designed to have the desired effect). Second, the hydraulic switch idea ignores the crucial issue of where the water comes from: i.e. the influence of surface-to-bed drainage and basal melting over surge cycles. Enthalpy balance theory (Benn et al. 2019) addresses these problems and unites surge mechanisms in a single framework, which potentially provides a more complete explanation of the sequence of events observed on Shisper Glacier. In particular, the exponential speed-up of Shisper in 2018 is consistent with frictional heating - velocity feedbacks, and the subsequent peaks and troughs in velocity appear to reflect competition between surface water inputs and drainage against a background of high frictional heating, all of these being associated with fluctuations in N. Overall, the pattern of surge evolution is remarkably similar to that observed during a recent surge of Morsnevbreen, Svalbard (paper cited below), and possibly for the same reasons. I suggest the authors investigate this possibility and weave it into their discussion of the Shisper Glacier surge.

We propose to combine sections "6.2 Surge trigger" and "6.6 Outlook on surge behavior", into one "6.5 Outlook on surge behavior" section (see proposed section below). We now include a discussion of how the enthalpy / mass budget model can help interpret certain aspect of the surge and the added value of the generalized sliding relationship to explain surge behavior. We still answer to each of the reviewers point independently as to motivate the ideas presented in the re-written section.

Out-dated binary choice of surge mechanism:

It is difficult to write a paper that describes surging without acknowledging predominant theories in the field, which is what we do. The enthalpy model is relatively recent (late 2019), and while it can produce an explanation as to why glaciers surge, the specific trigger mechanism is lacking, because of its lumped nature. In this paper, we focused the discussion on trigger and surge evolution, effect that are not captured by the lumped model

Dismissal of thermal mechanism:
The mechanism that we dismiss here is the 'thermal switch' from a frozen bed to a bed at the pressure melting point as a trigger for the surge. This mechanism is still popular and invoked in the recent literature (e.g., Farinotti et al., 2020). In that sense we consider shear heating a potential source of meltwater, not a thermal switch. The contribution of the shear heating on meltwater production was indeed not discussed but has now been added.

Lack of existence of hydraulic switch:
We are not aware of studies suggesting that the hydraulic switch is problematic. Kamb et al. 1987, indeed suggests a hydraulic switch to terminate the surge, although that paper is a model of subglacial drainage only and is not cited in our paper. The paper that we refer to is

Kamb et al. (1985). In that paper, the authors advance the idea of the drainage system closing at the end of the melt season leading to an increase in water pressure and thus velocities.

There is field evidence for such switch, which is not in essence "spontaneous" as suggested, but gradual (that takes weeks to month). When water supply tappers off, the channelized drainage system closes and leads to an increase in water pressures at the bed. This mechanism is illustrated in Fig.3 of Iken and Truffer (1997), where velocities show a peak in the summer followed by a decrease until the early winter, then rise again until the next summer. That rise is not associated with surface meltwater production. Our observations similarly point to a mechanism that leads to a Fall speed-up. We suspect that, in accordance with Kamb et al. (1985)'s idea, it could be due to the closing of an efficient, channelized drainage system. We think that this mechanism took place at Shisper prior to the surge, as shown in the data, and could have initiated the slow onset of the surge.

Origin of the water:

The Fall speed-up idea is centered on the fact that less water exits the glacier as the efficient drainage system shuts down. The temperature reanalysis data shows that the mean surface temperature at the elevation of the main trunk remains above 0degC at the time of the Fall speed-ups and the slow surge onset. We thus expect a significant amount of meltwater to be produced at the bed in addition to basal melt due to geothermal heat flux and shear heating.

Note that we updated Fig. 5 to now show the mean daily temperature at the average elevation of the main trunk. Previously it was showing the maximum daily temperature at the mean elevation of the re-analysis grid, which was >4000m a.s.l.
ADD NEW FIG 5

Basal melting is a somewhat continuous process and can initiate a gradual increase in sliding velocity, but it is unlikely to trigger sharp changes in velocities. With a gradual increase in water input, the subglacial drainage network would adjust to new water input and prevent a step change in effective pressures and sliding velocities (See Flowers 2008, Hydrological Processes, DOI: 10.1002/hyp.7095). It is also important to note that if the background water input remains too high, an efficient drainage network would be sustained, and surface velocities changes would be reduced. Here, we show a clear correlation between hydraulic events and sliding velocities.

In quantifying the contribution of basal melt, before the surge onset, it is small fraction of the water input (See added figure at the end of the answer). During the surge, it may become more significant and is likely to contribute to sustaining high velocities. However, all the large velocities changes observed here can be linked directly to hydraulic events similar to those predating the surge. Either a Fall or spring speed-ups or the formation of the lake.

Therefore, we believe that the enthalpy model can be essential in understanding the gradual increase in spring speed up amplitude and the fact that the glacier reaches an unstable equilibrium. In light of the data presented here and other observations of surges and the generalized sliding law, we believe that our interpretation that changes in N associated with hydraulic events drive the surge and its fluctuations holds. The basal melt production is likely to contribute to the surge amplitude, but we do not believe there is enough evidence here to suggest that it drives surge evolution.

Enthalpy / mass budget model integration with sliding rel to explain surging:

We have added this in the new proposed paragraph in the discussion.

Figure 7 shows a conceptual representation of σmax alongside the observed velocity fluctuations of Shisper Glacier. If the general sliding law is reframed with N as an explicit variable, it should be possible to replace this conceptual curve with estimates of N - potentially a much more interesting and valuable outcome. Figure 8 shows that the relationship between driving stress and velocity is extremely noisy, and this noise is attributed in the Discussion largely in terms of variations in N. If this is true, we should expect to see some structure in calculated values of N in time and space. Uncertainties will of course persist (e.g. variation in a and b as defined above; errors in the ice thickness derived from the Farinotti models), but if the authors' arguments are correct, some structure should be apparent. I urge the authors to explore this possibility in detail.

As stated earlier, we cannot reframe the sliding relationship as an explicit function of N, which makes inferences of N difficult. We believe it is best to leave such quantification to further studies with better data. As pointed out by the other reviewers the uncertainty surrounding ice thickness and surface elevation data makes variable parameter quantification difficult. The importance of N, is however, now more detailed in the reframed discussion.

The existing discussion of the importance of N in surging is not new. N is an integral component of the hard-bed and soft-bed sliding laws from which the 'generalised law' is derived, so it cannot be claimed that the generalised law 'highlights the importance of effective pressure'. However, if the authors can pull estimates of N from their excellent dataset and show how it varies during quiescence and surge, that would give the paper something really unique and valuable.

It is not new indeed, as we acknowledge. What is new is a unified framework that allows to show it more clearly. Changes in the text proposed for section 2, and the new discussion

paragraph, make the role of N clearer. This also departs from, for example, the enthalpy / mass budget approach that does not explicitly account for N. It is implicitly present in the water budget though. In many surge studies, the emphasis is often put onto the driving stress increase / decrease to explain surge onset / termination, as short-term mechanisms. What we find here is that these, similarly to the enthalpy model, are likely necessary to put the glacier in a surge-likely stage, but the short-term events responsible for the surge are probably associated with abrupt changes in N.

Proposed changes to the manuscript: combining sub-sections 6.2 Surge trigger and 6.6 Outlook on surge behavior. The new section is the following.

[revised manuscript text omitted]

*Figure 2: TO BE ADDED TO SUPPLEMENT. Calculation of ice melt in ice-thickness equivalent over a given area calculated as a function of shear heating with the high and low bound we find for the generalized sliding relationship and the expected surface melt for surface temperature of 2 and 4 degrees Celsius. These temperatures are what we observe in the re-analysis data around the times of Fall speed-ups for the main trunk zone. We used a degree day factor of 4.5, which is realistic for the Chinese Karakoram (Zhang et al. 2006, Annals of Glaciology)*

[Figure]

*Figure 3: reproduction of Fig. 8 with the model 1 of Farinotti et al. (2019). The parameters for drawing the sliding relationship are the same as used in the main manuscript.*

[Figure]

*Figure 4: reproduction of Fig. 8 with the model 2 of Farinotti et al. (2019). The parameters for drawing the sliding relationship are the same as used in the main manuscript.*

[Figure]

*Figure 5: reproduction of Fig. 8 with the model 3 of Farinotti et al. (2019). The parameters for drawing the sliding relationship are the same as used in the main manuscript.*

---

## Referee Report (RR1)

**Review of revisions made for "Generalized sliding law applied to the surge dynamics of Shisper Glacier and constrained by timeseries correlation of optical satellite images"**

I have read the revised manuscript, edits, as well as the authors response to my comments and the other reviewers. In the revised draft, some nice improvements are made in the discussion about the surge, however the main criticisms with regards to the bed friction are largely unaddressed. Much of the manuscript is publishable and interesting, but to reiterate my comments and that of another reviewer, the paper is trying to do a lot, there is plenty of material and nice data to discuss about the surge dynamics, but it gets overwhelmed especially in the discussion about friction changes which are poorly constrained. The difficulty of investigating friction relationships stems from the fact friction sliding velocity are difficult to estimate, even with good data and assumptions. In this regard, not even a relatively low bar is met and accordingly very few conclusions can be made in this aspect of the data analysis. Further modification will be necessary before publication.

I address some of the friction related comments here and then I propose some modifications. I will leave the other aspects (i.e. the surge behavior) to the other reviewers since Doug Benn is far more knowledgeable than I with regards to surge dynamics.

**Author Comment:**

We agree that the uncertainties are large and difficult to estimate, hence our choice of a qualitative interpretation. We tested different bed and surface topography models, the results show that the signal is large enough that the main conclusions hold regardless of the bed elevation model or surface elevation model used. The proposed changes should make this point clearer in the manuscript. The reviewer's statement about the methods presented in Farinotti et al. (2019) is not exactly correct. Two out of the three models (models 1-3 in Farinotti & 2019) include a parametrization of basal sliding while assuming simplified physics and the 3rd is mostly empirical. These models have been applied to marine terminating glaciers and their validity is thus not limited to a deformation-dominated flow scenario.

**My Comment:**

The uncertainty is not quantified in the manuscript and a qualitative understanding still requires knowing that your signal is above the noise, which is not demonstrated here. The interpretation is also not qualitative, rate-weakening and parameter bounds are quantitative aspects of the data. As far, as the Farinotti goes, you are correct that several of the parameterizations include sliding. However, these inversions schemes are designed to calculate world-wide ice volumes using simplified inversions tuned on a regional basis and are thus subject to high uncertainty for individual glaciers.

**Author Comment**

Using the driving stress to estimate the basal shear stress will lead to an overestimation of basal shear stress (see Minchew 2016, Thogersen 2019, etc.). That means that if we can show rate independent or rate-weakening behavior, that would only be further confirmed by a better quantification of basal shear stress.

**My Comment:**

The friction field is not quantified to the degree needed to claim rate-weakening behavior. Further, the driving stress is not a high-end member estimate basal shear stress. This is especially true during large

transient changes. Here, the global force balance must be maintained. This means regions where the friction is reduced will be accommodated by regions or lateral margins where the friction is increased through stress transfer. This is why a more sophisticated inversion for the basal shear stress that includes the full momentum balance is required to look at friction variations during a glacier surge.

The authors are clearly is in favor of integrating the unified friction theory and surge observations, here are some changes I would suggest for the manuscript to be publishable close to its current form:

Lines are in reference to the track changes document:

Lines: 475-495, 514-516, 608-610, Table 3 – Any discussion where it is claimed a range of parameters is found by bracketing the scatter. With the three sources of uncertainty which are unquantitified and also have the potential to be huge, this is not demonstrated, not even qualitatively as the authors claim. These lines and related discussion should be removed. As well as the last paragraph of the conclusion.

Lines: 455-460 – This should be upfront after the second sentence. It also needs to directly acknowledge all three sources of uncertainty, and roughly how they could influence the friction field, the fact that they have the potential to be large, but you are proceeding because you are mainly using this as a proof of concept.

Lines: 453-455 – Not necessarily true – see comments above.

An idea for discussion that incorporates the unified sliding framework: a discussion on how areas with high driving stress prior to the surge seem to be the regions that accelerate the most during the surge. This is interesting, and less hampered by uncertainty. You could easily relate this to your bed friction model, i.e. what are the conditions that need to be satisfied for this to occur? This discussion is made simpler in your unified framework. Keep in mind a friction change is likely associated transients during the surge which would not be captured with the driving stress approximation.

Section 6.3 "Towards a unified glacier sliding relationship": Could put the last three sentences of the conclusions right after 475, which would emphasize the section heading.

Section 6.4 - I would try to shorten the content into 1 paragraph and stick into section 6.3. The paper is already very long, and a long discussion about the general context of friction relationships and other peoples friction relationships is not needed here.

Title: Would possibly revisit depending on the revisions.

---

## Referee Report (RR2)

**Review of the article entitled "Generalized sliding law applied to the surge dynamics of Shisper Glacier and constrained by timeseries correlation of optical satellite images"**

**1   General comments**

This is an interesting and well documented study that attempts to define and apply a generalised sliding law to remotely sensed velocity of a glacier surge. The paper contains a wealth of data and present very clearly both the thought process and history behind the proposed sliding law and the specifics of the treatment of the remotely sensed velocities. That is very commendable but makes in the end for quite a long paper in which it feels that some of the messages are a bit diluted. This has already been a comment from the first round of review and the authors decided to maintain he structure of their paper as they feel that the sliding law part would not hold as a single study. My opinion on that is that splitting the paper would allow to get more in the details of the processes and limitations of the method while making the overall message of both part of the study clearer. If the author still decide to stick with the present structure I would urge them to review the title of their study as it seems that this sliding law is not really applied to the surge dynamics itself but more that the surge dynamics are used to infer the validity of said friction law.

I find the remote sensing part of the manuscript along with the description of the surge mechanisms a great addition to the literature, and the clarifications of the author following recommendations of the preceding round of review are satisfying in my opinion.

I have more issues with the part pertaining to the sliding law and its application.

- I am not completely convinced that the comments of the author relating to the shortcoming of the method are completely clear in the manuscript. The authors made a great effort in explaining the reasoning behind their choice of bedrock geometry and approximations used for the basal shear stress in the review answer but it feels that those are not that clear in the manuscript. Perhaps adding a section

stating more clearly the goals of the study and the reasoning behind the chosen approximations might clarify the message here.

- D. Benn commented during the first round on the shadowing of the effective pressure in the chosen formulation of the friction law. The authors made the reason of this choice quite clear in the manuscript and I can see the use of the friction law as it is but in my opinion it presents some limitations that should be more clearly presented. In its current form, it seems difficult to apply this law in a coupled approach where effective pressure would be computed, in that sens it seems that the term of generalised might not be well suited and that those limitations should be clarified.

- Regarding the analysis of the driving stress vs. excess velocity plots, there are a few points on which I would need some clarification.

  – The excess velocity itself is computed using the quiescent phase velocity as a reference, I wonder why the authors elected to use this velocity rather than the mean winter velocity. It feels to me that the mean quiescence velocity is roughly doubled the mean winter velocity and that difference would mostly be due increase sliding during summer.

  – Figure 8 and associated sup figures only use a division between quiescence and surge periods, I wonder why the different phases of the surge were not considered here and if it could yield more information.

  – On the design of Figure 8 I wonder if there could be some improvements to give a better idea of the fit of the curve. I would like to see the "Whole timeseries" panel without the over-imposed sliding laws to get a more unbiased view of the data points. It might also be useful to use the quiescence phase panels to zoom-in on the lower velocities and have a better idea of the fit of both curves in this range of velocities.

- On the discussion regarding the initiation of the surge I would like to get more information on the proposed mechanism, the author state that the perturbation is caused by glacier hydrology but the surge acceleration is happening after the last fall acceleration when we would expect the hydraulic system to be in a rather dormant state. I also think that missing to the discussion between surface and shear heating generated water is the difference in the temporal production of both those sources, one being seasonal and the other evolving more smoothly through time.

**2  Specific comments**

The version of the supplementary material that as been uploaded is a track changed version and might then not be the final version, however, I noted a few issues with that. In the answer to reviewers, the authors state that they added "a figure in the sup mat showing the different bed elevations resulting from the 3 models and the composite bed elevation." I could not find this figure and I think it would be a nice addition to the paper.

- The first introduction of the supplementary material seems to miss a few words but it can also be issues with the track changed format.

- There is no reference to the lake volume presented here from the text.

- Figure S4.2 might be misleading, the seasonality of the surface melt should be emphasised here, as an example, during the surge initiation in the middle of winter, the shear heating melt would actually be the largest source of water for the glacier.

- In Figure S4.3 and following, the caption should be placed as in the main manuscript.

Bellow is a list of more specific and technical comments throughout the manuscript given with line numbers:

- Line 63: Typically has an extra "l".

- Line 76: Isn't it "as well as"?

- Line 196: ortho-rectification is misspelled.

- Line 234: On this line and following shouldn't the window be $w$?

- Line 248: This sentence is not very clear to me, I would suggest: "When a velocity map overlaps with the preceding one, we only keep the newer image in the overlapping period.".

- Line 263: The last reference to figure 4 is missing its panel (e).

- Figure 5: If possible it would be nice to add some kind of hashing for the periods in which the confidence in the data is lower.

- Line 288: The altitude taken for the temperature given here is different than the one stated in the caption of Figure 5.

- Line 327: It seems that the two sections on mass balance (5.3 and 5.4) could be merged together.

- Line 358: Not sure why this comment on velocity is in the mass balance section.

- Line 364: The comparison to Arolla glacier does not make a lot of sense to me.

- Figure 7: The grid in panel (a) should be pushed to the background

---

## Author Response (AR2)

Answer to reviewers "Generalized sliding law applied to the surge dynamics of Shisper Glacier and constrained by timeseries correlation of optical satellite images" by Beaud et al."

We would like to thank the two reviewers and the editor for their feedback on the manuscript and giving us the opportunity to revise and resubmit it for further consideration. The main point of concern was related to the uncertainty surrounding the estimation of resistive stresses to assess the sliding relationship. We have addressed this concern in two main ways:

(1) by interpreting the driving stress as an indirect corroboration of the sliding relationship, rather than the net sum of resistive stresses and

(2) by removing the numbers associated with the sliding relationship parameters.

Throughout the paper we have updated the idea of an assessment of the sliding relationship to state that we use the surge dataset to substantiate the sliding relationship. We believe that helps make it clearer that the analysis is qualitative only.

To remove the uncertainty associated with using driving stress as a proxy for the net sum of resistive stresses, we rephrased our argument as follows. Considering the force balance acting on a glacier, if the velocity increases significantly while the driving stress remains relatively steady, the resistive stress must decrease to accommodate for the velocity increase. We thus use the driving stress to make a qualitative argument rather than an estimation of resistive stress. Section 6.3 (Towards a unified glacier sliding relationship) has been rewritten to reflect these changes.

In section 6.3 (Implications for glacier sliding), we removed the assessment of sliding parameters and the discussion about their possible values and range. These statements were also removed from the conclusion in accordance with the comments from Reviewer #1.

Both reviewer suggested a title update and we propose to change from: Generalized sliding law applied to the surge dynamics of Shisper Glacier and constrained by timeseries correlation of optical satellite images

**То**

Surge dynamics of Shisper Glacier revealed by time series correlation of optical satellite images and their utility to substantiate a generalized sliding law

Our detailed answers and manuscript modifications (green) are detailed below, in line with the reviewers' comments (black). Note that the comments from Reviewer #1 come as a response from our answers from the first round of revision which are still present.

**Reviewer #1**

Review of revisions made for "Generalized sliding law applied to the surge dynamics of Shisper Glacier and constrained by timeseries correlation of optical satellite images"

I have read the revised manuscript, edits, as well as the authors response to my comments and the other reviewers. In the revised draft, some nice improvements are made in the discussion about the surge, however the main criticisms with regards to the bed friction are largely unaddressed. Much of the manuscript is publishable and interesting, but to reiterate my comments and that of another reviewer, the paper is trying to do a lot, there is plenty of material and nice data to discuss about the surge dynamics, but it gets overwhelmed especially in the discussion about friction changes which are poorly constrained. The difficulty of investigating friction relationships stems from the fact friction sliding velocity are difficult to estimate, even with good data and assumptions. In this regard, not even a relatively low bar is met and accordingly very few conclusions can be made in this aspect of the data analysis. Further modification will be necessary before publication. I address some of the friction related comments here and then I propose some modifications. I will leave the other aspects (i.e. the surge behavior) to the other reviewers since Doug Benn is far more knowledgeable than I with regards to surge dynamics.

**Author Comment:**

We agree that the uncertainties are large and difficult to estimate, hence our choice of a qualitative interpretation. We tested different bed and surface topography models, the results show that the signal is large enough that the main conclusions hold regardless of the bed elevation model or surface elevation model used. The proposed changes should make this point clearer in the manuscript. The reviewer's statement about the methods presented in Farinotti et al. (2019) is not exactly correct. Two out of the three models (models 1-3 in Farinotti & 2019) include a parametrization of basal sliding while assuming simplified physics and the 3rd is mostly empirical. These models have been applied to marine terminating glaciers and their validity is thus not limited to a deformation-dominated flow scenario.

**My Comment:**

The uncertainty is not quantified in the manuscript and a qualitative understanding still requires knowing that your signal is above the noise, which is not demonstrated here. The interpretation is also not qualitative, rate-weakening and parameter bounds are quantitative aspects of the data. As far, as the Farinotti goes, you are correct that several of the parameterizations include sliding. However, these inversions schemes are designed to calculate world-wide ice volumes using simplified inversions tuned on a regional basis and are thus subject to high uncertainty for individual glaciers.

**Author Comment**

Using the driving stress to estimate the basal shear stress will lead to an overestimation of basal shear stress (see Minchew 2016, Thogersen 2019, etc.). That means that if we can show rate independent or rate-weakening behavior, that would only be further confirmed by a better quantification of basal shear stress.

**My Comment:**

The friction field is not quantified to the degree needed to claim rate-weakening behavior. Further, the driving stress is not a high-end member estimate basal shear stress. This is especially true during large transient changes. Here, the global force balance must be maintained. This means regions where the friction is reduced will be accommodated by regions or lateral margins where the friction is increased through stress transfer. This is why a more sophisticated inversion for the basal shear stress that includes the full momentum balance is required to look at friction variations during a glacier surge.

We have removed our estimations of resistive stress and sliding relationship parameters, and reframed the argument. The global force balance indeed needs to be maintained. Since the driving stress overall decreases throughout the surge (Figure 9), at the exception of the terminus, the resistance must weaken during that time to accommodate the velocity increase. In addition, the importance of longitudinal and lateral stress coupling in the ice is enhanced as velocities increase (Blatter, 1995; Pattyn, 2002; Pattyn et al., 2008). This reduces the importance of the resistance at the bed and valley walls, further validating our findings.

This statement about the driving stress is generally true, however it is 1) not specifically what we were saying, and 2) only valid for relatively low sliding velocities. Our argument is specific to large sliding velocities, not in all flow conditions. This is backed up by several lines of evidence we could find in the literature where sliding speeds in excess of 0.3-0.4 m/day concur with driving stress in excess of the resistive stress. (Habermann & al 2013; Valot& al 2017; Hoffman and Price 2014; Minchew & al 2016; Thogersen & al, 2019). It is, in fact, well established that longitudinal stress gradients are essential to maintain the force balance at high sliding velocities, not the basal or wall resistive stresses, otherwise the shallow ice approximation would be valid for steep and fast sliding glaciers.

The authors are clearly is in favor of integrating the unified friction theory and surge observations, here are some changes I would suggest for the manuscript to be publishable close to its current form: Lines are in reference to the track changes document:

Lines: 475-495, 514-516, 608-610, Table 3 – Any discussion where it is claimed a range of parameters is found by bracketing the scatter. With the three sources of uncertainty which are unquantitified and also have the potential to be huge, this is not

demonstrated, not even qualitatively as the authors claim. These lines and related discussion should be removed. As well as the last paragraph of the conclusion.

**We have removed all quantifications of sliding relationship parameters.**

Lines: 455-460 – This should be upfront after the second sentence. It also needs to directly acknowledge all three sources of uncertainty, and roughly how they could influence the friction field, the fact that they have the potential to be large, but you are proceeding because you are mainly using this as a proof of concept.

We have re-framed the argument using the force balance and we now focus on driving stress rather than resistive stress. If velocity increases significantly while the driving stress doesn't, the net resistance must decrease. The uncertainty in the data presented is thus now only around the driving stress, more specifically changes therein. Here net changes in driving stress during the surge are captured by the changes between the DEMs, the ice thickness estimation is only important for absolute values of driving stresses.

We believe the reframing addresses the reviewer's comment as it makes our qualitative goal more clear.

Lines: 453-455 – Not necessarily true – see comments above.

We have removed the notion that the driving stress is used as a proxy for resistive stresses here.

Changes made at I.445-451 of track changed manuscript:

From:

The mean quiescence velocity (uquiesc) is calculated by time averaging the entire velocity field for each velocity map until the surge onset in November 2017. We choose to use the driving stress ( $\tau$ d) as an approximation for the basal stress ( $\tau$ b). While this assumption is coarse, it ensures that we overestimate the basal stress at large sliding velocities (see Minchew et al., 2016; Thøgersen et al., 2019), making it more difficult to observe rate-independent or rate-weakening behavior. We then plot the relationship  $\tau$ b versus uex for our dataset (Fig. 8). Equation 6 enables us to identify 4 free parameters ut, omax, p q that we can tune to bracket the data.

to

The mean quiescence velocity (uquiesc) is calculated by time averaging the entire velocity field for each velocity map until the surge onset in November 2017. We expect this choice to lead an overestimation of the mean quiescence velocity as it encompasses the enhanced speed-ups of 2015 and 2016. This underestimation of the sliding signal renders our assessment more conservative.

An idea for discussion that incorporates the unified sliding framework: a discussion on how areas with high driving stress prior to the surge seem to be the regions that accelerate the most during the surge. This is interesting, and less hampered by uncertainty. You could easily relate this to your bed friction model, i.e. what are the conditions that need to be satisfied for this to occur? This discussion is made simpler in your unified framework. Keep in mind a friction change is likely associated transients during the surge which would not be captured with the driving stress approximation.

We were not able to address this comment as it is unclear to us how the reviewer came to that conclusion. The area the most active is between km 10 and 12, yet it is where driving stresses are relatively low (Fig. 9). The highest driving stresses are observed at the ice fall which sees the least acceleration during the surge compared to other areas of the glacier.

Section 6.3 "Towards a unified glacier sliding relationship": Could put the last three sentences of the conclusions right after 475, which would emphasize the section heading.

A similar statement added in the last paragraph as the section was significantly rewritten.

Section 6.4 – I would try to shorten the content into 1 paragraph and stick into section 6.3. The paper is already very long, and a long discussion about the general context of friction relationships and other peoples friction relationships is not needed here.

Putting our research into the context of previous work and showing how it fits together appears to us as an important part of developing a generalized sliding relationship. The evidence we present here is mostly substantial, hence the necessity of building on and using the context from previous studies.

Title: Would possibly revisit depending on the revisions.

Title changed. See introductory comment.

**Reviewer #2**

Review of the article entitled Generalized sliding law applied to the surge dynamics of Shisper Glacier and constrained by timeseries correlation of optical satellite images

**1 General comments**

This is an interesting and well documented study that attempts to deffine and apply a generalised sliding law to remotely sensed velocity of a glacier surge. The paper contains a wealth of data and present very clearly both the thought process and history behind the proposed sliding law and the specifics of the treatment of the remotely sensed velocities. That is very commendable but makes in the end for quite a long paper in which it feels that some of the messages are a bit diluted. This has already been a comment from the first round of review and the authors decided to maintain the structure of their paper as they feel that the sliding law part would not hold as a single study. My opinion on that is that splitting the paper would allow to get more in the details of the processes and limitations of the method while making the overall message of both part of the study clearer. If the author still decide to stick with the present structure I would urge them to review the title of their study as it seems that this sliding law is not really applied to the surge dynamics itself but more that the surge dynamics are used to infer the validity of said friction law.

I find the remote sensing part of the manuscript along with the description of the surge mechanisms a great addition to the literature, and the clarifications of the author following recommendations of the preceding round of review are satisfying in my opinion.

**Thank you for your overall positive assessment of our revised manuscript.**

I have more issues with the part pertaining to the sliding law and its application.

• I am not completely convinced that the comments of the author relating to the shortcoming of the method are completely clear in the manuscript. The authors made a great effort in explaining the reasoning behind their choice of bedrock geometry and approximations used for the basal shear stress in the review answer but it feels that those are not that clear in the manuscript. Perhaps adding a section stating more clearly the goals of the study and the reasoning behind the chosen approximations might clarify the message here.

As suggested by reviewer #1, we have removed numbers associated with parameters and reframed the way we make our arguments in a clearly qualitative manner. In the

process, we have removed the explanations for the aforementioned approximations as the driving stress is only indirectly used to infer rate-independent stress. We believe that these changes simplify the reasoning as the justifications to use driving stress as a proxy for resistive stresses are not necessary anymore.

 D. Benn commented during the first round on the shadowing of the effective pressure in the chosen formulation of the friction law. The authors made the reason of this choice quite clear in the manuscript and I can see the use of the friction law as it is but in my opinion it presents some limitations that should be more clearly presented. In its current form, it seems difficult to apply this law in a coupled approach where effective pressure would be computed, in that sens it seems that the term of generalised might not be well suited and that those limitations should be clarified.

**Added text Just before S.3:**

"This highlights the variables of interest to use the generalized relationship in data inversion efforts or to refine rate-and-state approaches (e.g. Thøgersen et al., 2019). It can complicate the use of the relationship in numerical models where the direct use of N might be necessary, although existing models that readily solve for water pressure, and thus calculate N, will already have chosen a type of substrate as dictated by water flow in different media (see Flowers, 2015, for a review)."

The law is generalized in the sense that it can be applied to any glacier bed. As mentioned in the answer to D. Benn's comment, we did not make either of the existing laws more complicated than they already were. Rather, we show that the generalized formulation reduces to the previously published individual sliding laws under appropriate assumptions. Numerical models computing N already have to make a choice in term of substrate to determine which equations to use (till or hard bed), and will thus be able to choose the appropriate expression for the threshold velocity and maximum resistive stress.

- Regarding the analysis of the driving stress vs. excess velocity plots, there are a few points on which I would need some clarification.
  - The excess velocity itself is computed using the quiescent phase velocity as a reference, I wonder why the authors elected to use this velocity rather than the mean winter velocity. It feels to me that the mean quiescence velocity is roughly doubled the mean winter velocity and that difference would mostly be due increase sliding during summer.

**Added L439-451 of the track changed manuscript:**

We expect this choice to lead to an overestimation of the mean quiescence velocity as it encompasses the enhanced speed-ups of 2015 and 2016, meaning the excess velocity is likely underestimated. This underestimation of the sliding signal renders our assessment more conservative. - Figure 8 and associated sup figures only use a division between quiescence and surge periods, I wonder why the different phases of the surge were not considered here and if it could yield more information.

We explored ways to plot the data with its temporal evolution, but found that the interpretation was too challenging, especially since we lack evolving DEM throughout the surge over the whole glacier. We are also aware of uncertainty in bed data and wanted to ensure we did not over-interpret the data.

Added I. 458-459 of the tracked changes manuscript: We chose to lump the different phases of the surge together because we lack data on surface changes at a required temporal resolution during the event.

- On the design of Figure 8 I wonder if there could be some improvements to give a better idea of the fit of the curve. I would like to see the Whole timeseries panel without the over-imposed sliding laws to get a more unbiased view of the data points. It might also be useful to use the quiescence phase panels to zoom-in on the lower velocities and have a better idea of the fit of both curves in this range of velocities.

We removed the plotted relationship in the panel showing the whole timeseries to remove that possible bias. We kept the plotted lines in the other panels as we believe it is useful for the interpretation, but we made the line thinner and color lighter to make them less important.

As asked by the other reviewer, we also removed the idea of "fit" and associated legend and table with numbers but kept the lines to show the types of behavior.

We prefer to keep all panels with the same X-axis for the velocities. Zooming in would indeed give a better idea of the trend during the quiescence, but since we focus on the relatively large velocities here, we believe it is more appropriate for the current story. The part of the data set plotted in the supplement in log-log space in figure S4.9.

 On the discussion regarding the initiation of the surge I would like to get more information on the proposed mechanism, the author state that the perturbation is caused by glacier hydrology but the surge acceleration is happening after the last fall acceleration when we would expect the hydraulic system to be in a rather dormant state. I also think that missing to the discussion between surface and shear heating generated water is the difference in the temporal production of both those sources, one being seasonal and the other evolving more smoothly through time.

This has been clarified.

Also, as stated in the manuscript the surge onset is in November which coincide with the observed Fall speed-ups, not later.

On I.549-568 of the tracked-changed manuscript the text now reads:

Once the glacier is in an unstable equilibrium any perturbation can start the surge. For Shisper, we infer that that perturbation is caused by surface melt-dominated hydrology due to its periodic nature, rather than by a relatively steady meltwater production due to shear heating. Our data shows a pre-surge history of both Fall and spring speed-ups, that appears to be linked to the glacier's hydrology (Fig. 5), and thereafter the surge dynamics seems modulated by hydraulic events. Spring speed-ups are typically explained by an increase in water input overwhelming a mostly distributed subglacial drainage system (e.g. Müller and Iken, 1973; Iken and Truffer, 1997). On the other hand, the Fall speed-up is typically explained by the closure of a channelized drainage system that leads an overall increase in water pressure although the input does not necessarily increase, and that mechanism has previously been proposed to explain surge initiation (e.g. Kamb, 1987; Abe and Furuya, 2015). The presence of Fall speed-ups prior to the surge and its slow initiation showing a coinciding timing suggest that said Fall speed-up is responsible for the surge onset. The first main phase of the surge is then triggered by the following spring speed-up.

**2 Specific comments**

The version of the supplementary material that as been uploaded is a track changed version and might then not be the final version, however, I noted a few issues with that. In the answer to reviewers, the authors state that they added a figure in the sup mat showing the different bed elevations resulting from the 3 models and the composite bed elevation. I could not find this figure and I think it would be a nice addition to the paper.

Thanks for pointing out that omission, the figure is back where it belongs.

- The first introduction of the supplementary material seems to miss a few words but it can also be issues with the track changed format. Fixed
- There is no reference to the lake volume presented here from the text. Fixed
- Figure S4.2 might be misleading, the seasonality of the surface melt should be emphasised here, as an example, during the surge initiation in the middle of winter, the shear heating melt would actually be the largest source of water for the glacier.

Clarification added I. 50-575 of the tracked-change manuscript:

If we estimate the basal shear heating over the current data set, we find that while it is non-negligible (Fig. S4.2), it produces significantly less melt than the expected surface melting if the mean daily temperature is above 2°Celsius.

**Now reads:**

If we estimate the basal shear heating over the current data set, we find that while it is non-negligible (Fig. S4.2), it produces significantly less melt than the expected surface melting if the mean daily temperature is above 2°Celsius. This is commensurate with the surface temperature at the time of the surge onset in the main trunk.

The temperature of 2degC chosen to estimate surface melt in Fig. S4.2 is commensurate with the re-analysis data at the time of surge initiation (and other Fall speed-ups). This can also be seen in Fig. 5 where the surge onset corresponds to temperatures above 0degC. Our estimate thus show that at the time of onset, the surface melt is expected to be larger than basal melt in the main trunk of the glacier.

Reference to the temporal nature of the two forcing has been clarified in the caption as well.

 In Figure S4.3 and following, the caption should be placed as in the main manuscript. Fixed

Bellow is a list of more specific and technical comments throughout the manuscript given with line numbers:

- Line 63: Typically has an extra I Fixed
- Line 76: Isn't it as well as ? Fixed
- Line 196: ortho-recti cation is misspelled. Fixed
- Line 234: On this line and following shouldn't the window be w? Fixed
- Line 248: This sentence is not very clear to me, I would suggest: When a velocity map overlaps with the preceding one, we only keep the newer image in the overlapping period.

**We changed:**

"When a velocity map overlaps with the preceding one, we only keep the older velocity map for the time span between the two maps older images."

**to**

"When velocity maps overlap, we use the velocity map with the older starting image for the timespan of the overlap."

• Line 263: The last reference to figure 4 is missing its panel (e). Fixed

• Figure 5: If possible it would be nice to add some kind of hashing for the periods in which the confidence in the data is lower. The principal component analysis takes the signal to noise ratio into account to produce the final velocity maps.

• Line 288: The altitude taken for the temperature given here is different than the

one stated in the caption of Figure 5. Fixed

• Line 327: It seems that the two sections on mass balance (5.3 and 5.4) could be merged together. Merged.

References in comments but not in the manuscript:

Blatter, H., 1995. Velocity and stress fields in grounded glaciers: a simple algorithm for including deviatoric stress gradients. J. Glaciol. 41, 333–344.

Pattyn, F., 2002. Transient glacier response with a higher-order numerical ice-flow model. J. Glaciol. 48, 467–477.